# Analysis of a self-supporting shell concrete roof with nonlinear coupled evolutive material parameters

**Landry Djopkop Kouanang**[1,2]*, **Merlin Bodol Momha**[1], **Daniel Ambassa Zoa**[1,2],
**Jean Chills Amba**[1,2], **Joseph Nkongho Anyi**[1,3], **Robert Nzengwa**[1,2]

**1** Laboratory of Energy Materials Modeling and Methods (E3M), Douala, Cameroon, **2** National Higher Polytechnic School of Douala, Douala, Cameroon, **3** Higher Technical Teacher Training College, Kumba, Cameroon

* landriville35@gmail.com

## Abstract

In many concrete-design civil engineering constructions, structural analyses are performed through finite element methods on an ideal equivalent elastic homogeneous material. However, in some cases, the evolution of these structures is impacted by delay effects (creep, shrinkage, etc.) and hydration, which sometimes also create structural damage. In this work, we propose a design method that includes the thermochemical and hydromechanical (TCHM) behaviour of concrete materials. An experimental design was carried out on concrete samples cast under laboratory conditions to monitor strain. A finite element method was subsequently used to simulate the behaviour of the sample under drying conditions. The gradient development linked by a nonuniform moisture distribution in the thickness was established by solving the nonlinear partial differential drying equation with Mensi's diffusion law. The stress and displacement analysis was modelled by sixnodes (MT6) based on strain approximation with shell theory. The results indicated that considering the delayed effects associated with the mechanical change on the thickness variation produces an effect identical to that which would have been produced by an individual mechanical loading worth 183 times the value of the mechanical loading considered. The deformation was calculated using the finite element method. This method was successfully applied to a self-supporting concrete roof.

## 1. Nomenclature

Note: summation notation is used to define the parametrization and calculation of the shell

$(x, z) = (x^1, x^2, z)$ parametrization of the shell; $z$: coordinate through the thickness;

$(A^\alpha, A^3)$, $(A_\alpha, A_3)$ contravariant and covariant base on the midsurface;

$(G^\alpha, A^3)$, $(G_\alpha, A_3)$ contravariant and covariant base of the shell;

**Data availability statement:** All relevant data are within the paper and its Supporting Information files.

**Funding:** The author(s) received no specific funding for this work.;

**Competing interests:** The authors have declared that no competing interests exist.

$U(x, z) = U_\alpha(x, z)\, G^\alpha + U_3(x, z)\, A^3$: displacement vector;

$B_\alpha^\rho, B_{\rho\alpha}$: mixed and covariant components of the curvature tensor respectively;

$\delta_\alpha^\rho$: Kronecker's symbol;

$\mu_\alpha^\rho = (\delta_\alpha^\rho - z B_\alpha^\rho)$

$G_\alpha = \mu_\alpha^\rho A_\rho$

$G^\alpha = (\mu^{-1})_\rho^\alpha A^\rho$;

$\psi = \det(\mu_\alpha^\rho) \neq 0$ so that $\{G_\alpha, G_3\} = \{G_1, G_2, G_3\}$ constitutes a base;

$h$: shell thickness;

$\gamma_{\alpha\beta}(x) = -\frac{1}{2}\left(\mu_\alpha^\rho B_{\rho\beta} + \mu_\beta^\rho B_{\rho\alpha}\right)$: the section warping tensor;

$w(z)$: the transverse stretching distribution function;

$\bar{\Gamma}_{\beta\alpha}^\rho$: Christoffel symbol on the midsurface;

$\in^e$: elastic strain tensor;

$\in^{sh}$: shrinkage strain tensor;

$\in^{th}$: thermal strain tensor;

$\in^{cr}$: creep strain tensor;

$\in$: total strain tensor;

$d$: damage coefficient;

$\omega_L$: free water content (L/m³);

$\omega_h$: bound water (L/m³);

$\omega_0, \omega_f$: initial water content and moisture content in equilibrium with external hygrometry;

$\xi$: degree of hydration;

$h_c$: convective coefficient;

$\delta$: material drying velocity coefficient.

## 2. Introduction

Thin or thick concrete shell structures (arched dams, bridge caissons, domes, buried or raised tanks, etc.) crack, deform and sustain damage during their service time, as can be observed in classical concrete structures. These defaults are not always predictable because design calculations are often based only on the elastic behaviour of the material. Several researchers are trying to solve this problem by integrating the peculiarities of crackable concrete materials into the calculations. In [1] and [2], a solution to integrate some of these particularities is proposed through the introduction of superimposed elements, which is based on degenerate finite elements. Some additional specificities are considered in [3–5]. Notably, however, thickness variation, transverse shear stresses and hydration were not addressed in these previous works, yet dimensional variations due to hydration reactions and delay effects taking place within the material impact thickness. The material behaviour thus considered is not realistic. In [6], a modified compression field theory developed in the above papers was proposed to integrate transverse shear strains but still neglected thickness variation. Later, M. A. Polak and F.Veccicho [7] proposed the smeared rotary crack approach, which improved the calculation of transverse shear strains through a nonlinear process based on orthotropic material and the formulation developed in [3,4] and [5]. In [8–11], the authors proposed a model that considers cracks in

reinforced concrete due to traction, damage to the concrete due to compression, development of residual strains, slippage between the concrete and steel rods and plasticization of the steel rods. The transverse shear strains are calculated by the Reissner–Mindlin shell elements (with transverse shear). These models are invariant-thickness based.In [12] M. Huguet et al. developed a new nonlinear method to calculate stresses in a representative volume element delimited by two cracks in reinforced concrete subjected to cyclic loading, incorporating hypotheses on concrete damage, concrete cracking, bond sliding stress and steel performance. This model has substantial memory requirements and encounters numerical locking phenomena when the shell is thin. Other authors, A. Aili et al [13] presented two different approaches to predict delay strains. The first is a decoupled approach, which divides the delay strain into four components and predicts each of them as a function of several parameters, such as concrete strength. In the second approach, deferred strain is modelled as the viscoelastic response of the concrete to applied external loads and/or internal moisture stresses. The advantages and disadvantages of both methods are presented. It should be noted, however, that the authors consider the concrete material to be homogeneous, which is a strong assumption.

Cracks, damage and even failure of some concrete structures may also be caused by delay effects (shrinkage, creep, thermal effects or their interaction, etc.) [14] which always begin after hydration. Considering these phenomena during design calculations is necessary for structural safety. Several authors have studied global hydration models. Some depend on time directly; [15–17] and [18] and some indirectly (multiphase approach TCHM) [19–21] In view of the proposed expressions, we realize that some relationships do not consider the nonuniform variation in bound water. The model defined by Merlin, Bodol Momha, et al. [22] includes diffusion models proposed by [23,24] and seems more realistic. Some authors, [25–28] have implemented a model that considers delay phenomena in massive structures. D. Gawin, et al [29] also makes proposed a mathematical model for analysing the hygrothermal behaviour of concrete as a multiphase porous material at very high temperatures, considering material deterioration. The model's equations are then developed from the macroscopic mass, energy and linear momentum balances of the individual constituents. The ACI 209-R92 model, initially developed by Branson, D. E et al [30] and adopted by the AIT Committee 209 with some modifications, is an empirical model, that proposes a formula to determine shrinkage deformation and the creep coefficient. B4 is a model recommended by the RILEM TC-242-MDC committee and is based on the B3 model [31] development of which began in the 1970s. The model is based on the mathematical description of ten mechanisms affecting shrinkage and creep [32].The model is particularly recommended for detailed design and verification. Practitioners prefer the modified version of the GL2000 model [33] by [34], because the input parameters are few and are available at the design phase. The model also treats the case of post drying loading. The AASHTO 2012 model which was developed in [35], and published by the American Association of State Highways and Transportation Officials, is a modified version of [36] and [37]. The model does not distinguish autogenous shrinkage from drying shrinkage. In Appendix B of, [38] the model calculates a creep coefficient instead of the creep function. This model distinguishes proper creep and desiccation creep.

In view of previous works, several couplings have been proposed to explain the behaviour of concrete materials. Notably, there is an under representation of works that address a coupling that takes into account both delay effects and the hydration reaction and its effects within the material, with the aim of assessing the deformation of the structure over time. In this work, we consider the coupling of time-delay effects and the phenomenon of concrete hydration to assess displacements in the shell. A thermochemical and hydromechanical (TCHM) analysis was performed. The finite element analysis is based on shell elements [39–41] that consider transverse shear deformation and thickness variations.

Notably, the scope of this work lies in considering the material nonlinearity of concrete. This allows us to realistically calculate or predict the behaviour of the concrete structure over time.

The remainder of the paper is organized as follows: Section 3 presents of the analysis of a self-supporting shell steel-reinforced concrete roof subject to nonlinear coupled evolutive material parameters. Numerical implementation is developed in Section 4. In section 5 we present the results and discussion. Our conclusions are presented We in section 6.

## 3. Model presentation

Let us start by describing the different phenomena involved and the right global coupled relationships.

### 3.1. Concrete shrinkage tracking

As described above, to enhance the design of concrete structures, the hydration and drying phenomena should be integrated as defined here.

**3.1.1. Hydration phenomenon.** The degree of hydration is described by the Arrhenius law [22] as follows

$$\frac{d\xi}{dt} = \tilde{A}(\xi)\beta_h \left(\frac{\omega_L}{\omega_0}\right) \exp\left[-\frac{E_a}{RT}\right];$$

(1−1)

where $\tilde{A}$ denotes the chemical affinity, $E_a$ represents the activation energy, R represents the perfect gas constant, T represents the ambient temperature and $\beta_h$ is obtained from the isotherm curve desorption of the concrete. The time evolution of Young's modulus of the concrete depends on the hydration kinetics as follows [19].

$$E(\xi) = E_\infty \left(\frac{\xi - \xi_o}{1 - \xi_o}\right)^b$$

(1–2)

- $E_\infty$ is the ultimate Young's modulus for complete hydration [GPa];

- $\xi_o$ represents the degree of advancement corresponding to the mechanical percolation threshold;

- b is a parameter that depends mainly on the type of cement used. We retain the constant defined by Zreiki et al. [42] in their works on concrete $\xi_o = 0.223$ and b = 0.25.

**3.1.2. Drying phenomenon.** We use the following expressions proposed by [23,24] and Mensi et al for the determination of $\omega_L$ (free water content (l/m3)).

$$\frac{\partial \omega_L}{\partial t} = div\left(D_v(\omega_L)\nabla\omega_L\right).$$

(2)

$$D_v(\omega_L) = \bar{A}\exp(\bar{B}\omega_L)$$

(3)

- $D_v$ is the water diffusion coefficient

The parameter $\bar{B}$ can be taken as a constant: $\bar{B}$ = 0.05. The drying parameter $\bar{A}$ needs to be determined. The boundary conditions used are in the convective form,

$$h_c = \delta\left(2\omega_0 - \omega_f - \omega_L\right);$$

(4)

The normal flow at the exposed face (extrados) can be written as follows:

$$D_v(\omega_L)\frac{\partial\nabla\omega_L}{\partial n}.n = -\delta\left(2\omega_0 - \omega_f - \omega_L\right)\left(\omega_L - \omega_f\right).$$

(5)

The drying parameters $\bar{A}$ and $\delta$ are numerically adjusted after the resolution. These parameters are fit based on experimental data.

The loss of moisture during drying results in a contraction of the cement paste, as shown in the equation below:

$$\in^{dsh}(x, t) = p\omega_e(x, t);$$

(6)

where $\omega_e$ is the evaporable water. We assume that total shrinkage is the sum of drying and endogenous shrinkage, such that $\in^{sh} = \in^{esh} + \in^{dsh}$, and

$$\in^{esh}(x, t) = \kappa\Delta\omega_h(x, t)I.$$

(7)

where I is the identity matrix, $\Delta\omega_h$ is the gradient of bound water and $\kappa$ (autogenous shrinkage coefficient) and $p$ (drying shrinkage coefficient) are calibration parameters. We adjust them using experimental data.

### 3.2. Thermal behaviour

We approach thermal shrinkage by using the coefficient of thermal expansion $\alpha^{th} = 10^{-5}/{}^0C$ according to [43] as follows:

$$\in^{th} = \alpha^{th}\Delta T$$

(8)

where $\Delta T$ is the temperature variation. Given the small thickness of the shell, we assume that the temperature of the thickness remains constant.

### 3.3. Creep

We consider the creep function defined in [38] as follows.

**3.3.1. Self-creep.** Self-creep is expressed as follows:

$$\varphi_b(t, t_o) = \varphi_{bo}\frac{\sqrt{t-t_o}}{\sqrt{t-t_o} + \beta_{bc}}$$

(9)

where $t_o$ is the age (in days) when the shrinkage load (or deformation) is applied.

$$\varphi_{bo} = \begin{cases} 3.6/\ [f_{cm}(t_o)]^{0.37} & \text{for concretes with silica fume} \\ 1.4 & \text{for concretes without silica fume} \end{cases}$$

(10)

$$\beta_{bc} = \begin{cases} 0.3exp\{2.8f_{cm}(t_o)/f_{ck}\} & \text{for concretes with silica fume} \\ 0.4exp\{3.1f_{cm}(t_o)/f_{ck}\} & \text{for concretes without silica fume} \end{cases}$$

(11)

where $f_{cm}(t_o)$ is the average strength of the concrete at $t_o$, and $f_{ck}$ is the characteristic strength of the concrete at 28 days.

**3.3.2. Desiccation creep.** Desiccation creep is expressed as follows [38]:

$$\varphi_d(t, t_o) = \varphi_{do}[\epsilon_{cd}(t) - \epsilon_{cd}(t_o)]$$

(12)

$$\varphi_{do} = \begin{cases} 1000 & \text{for concretes with silica fume} \\ 3200 & \text{for concretes without silica fume} \end{cases}$$

(13)

where

$\epsilon_{cd}(t)$ is the desiccation shrinkage,

$$\in_{cd}(t) = \frac{K(f_{ck})\left[72\exp(-0,04f_{ck}) + 75 - RH\right](t - t_s)10^{-6}}{(t - t_s) + \beta_{cd}h_0^2}$$

$K(f_{ck}) = 18$ if $f_{ck} \leq 55$ *MPa*
$K(f_{ck}) = 30 - 0,21f_{ck}$ if $f_{ck} > 55$ *MPa*

$$\beta_{cd} = \begin{cases} 0.007 \text{ for concretes with silica fume} \\ 0.021 \text{ for concretes without silica fume} \end{cases}$$

$RH = $ *Relative humidity*;

The total creep function is written as

$$J(t, t_o) = \frac{1}{E_c(t_o)} + \frac{\varphi_b(t, t_o)}{E_{c28}} + \frac{\varphi_d(t, t_o)}{E_{c28}} = \frac{\exp\left\{-0.3s\left(1 - \sqrt{28/t_o}\right)\right\}}{E_{c28}} + \frac{\varphi_b(t, t_o)}{E_{c28}} + \frac{\varphi_d(t, t_o)}{E_{c28}}$$

$$\in^{cr} = \sigma_o J(t, t_o) \tag{14}$$

where $\sigma_o$ is imposed stress

### 3.4. Damage

We use the MAZARS model of behavior, which allows us to describe the rubber band-damage behaviour of the concrete. This model is 3D, isotropic and is based on a criterion of damage written in deformation dissymmetry traction and compression. The damage ratio proposed in [44], uses a scalar variable "d" that corresponds to the damage in the already homogenized directions; and the value is between [0,1].

$$d = 1 - \frac{(1 - A)Y_o}{Y} - A * \exp(-B(Y - Y_o)) \tag{15}$$

In this expression (Equation 15), the variables A and B allow us to reproduce the quasi-fragile behaviour of concrete under traction and the hammer-hardened behavior under compression. To represent the experimental results as well as possible, the following laws of evolution were selected for A and B (Equations 17 and 18).

$$Y_o = \varepsilon_{do} \text{ ò } [0.5 - 1.0] * 10^{-4}; \tag{16}$$

$$A = A_t\left(2r^2(1 - 2j) - r(1 - 4k)\right) + A_c\left(2r^2 - 3r + 1\right) \tag{17}$$

$$B = B_t r^2 + B_c\left(1 - r^2\right) \tag{18}$$

where the expression of r is as follows:

$$r = \frac{\sum_i \langle \tilde{\sigma}_i \rangle_+}{\sum_i |\tilde{\sigma}_i|} \tag{19}$$

Equations (17) and (18) a new variable $r$ which provides information about the state of stress. A value of $r = 1$ corresponds to the sector of tractions.

$$Y = \max\left\{\varepsilon_{do}, \max\left(\varepsilon_{eq}^{corr}\right)\right\} \tag{20}$$

The improvement in behaviour is that shearing is reached by the introduction of a new internal variable: $Y$ (defined by Equation (20)). This value corresponds to the maximum reached during the loading of the equivalent deformation. Its initial value $Y_o$ (defined by Equation (16)) is $\varepsilon_{do}$.

$$\varepsilon_{eq}^{corr} = \gamma\sqrt{\langle\varepsilon\rangle_+ : \langle\varepsilon\rangle_+} \tag{21}$$

With the $\varepsilon$ tensor of deformations, the positive part $<>_+$ is defined such that if $\varepsilon_i$ is the principal deformation in the direction i then:

$$\begin{cases} < \varepsilon_i >_+ = \varepsilon_i, & \text{si } \varepsilon_i \geq 0 \\ < \varepsilon_i >_+ = 0 & \text{si } \varepsilon_i < 0 \end{cases}$$

$$\begin{cases} \gamma = -\dfrac{\sqrt{\sum_i \langle\tilde{\sigma}_i^2\rangle_-}}{\sum_i \langle\tilde{\sigma}_i\rangle_-}; \text{if at least one effective stress is negative} \\ \gamma = 1 \text{ ; } otherwise \end{cases} \tag{22}$$

In the field of tractions, $A = A_t$ and the opposite is true in the area of cuts$A = A_c$. Generally, and$0.7 < A_t < 1$. $B$ depending on its value may correspond to a sudden drop in stress ($B > 10000$) or a preliminary phase of stress increase followed, after passing through a maximum, by a telatively rapid decrease. In the field of tractions, $B = B_t$ and opposite is true in the area of cuts$B = B_c$.Generally$1000 < B_c < 2000$, and$9000 < B_c < 21000$. These values are obtained via compression tests and tensile tests. We use the recommended value of $j = 0.7$ [44]. A value of j lower than 1 is very useful for modelling the effects of friction entering the concrete and the reinforcements in reinforced concrete structures because it induces residual shear stress.

### 3.5. Mechanical model

The boundary problem consists of finding the displacement field $U : (x, z) \in \overline{\Omega} \to \mathbb{R}^3$ which can be expressed componentwise indifferently in the G-base or the A-base as follows:

$$U = U_i(x, z)G^i = \left(\mu^{-1}\right)_{\rho}^{\alpha} U_{\alpha} A^{\rho} + U_3 A^3$$

with $U_{\alpha} = u_{\alpha} - z\left(\partial_{\alpha}u_3 + 2B_{\alpha}^{\rho}u_{\rho}\right) + z^2\left(B_{\alpha}^{\rho}B_{\rho}^{\tau}u_{\tau} + B_{\alpha}^{\rho}\partial_{\rho}u_3\right), \quad U_3 = u_3 + q(x, z)$ [41].

We are interested in the particular case in which the stretching displacement, where $w$ is an arbitrary nonconstant function or a polynomial of any degree that is zero at the midsurface, and $\bar{q}$ is a function that depends only on x, i.e., is defined on a plane. Let $v$ exist in $U_{ad}$, because $w$ is not constant in z, and the boundary condition of the displacement is$\eta_1 = \eta_2 = \eta_3 = \bar{q} = \partial_{\alpha}\eta_3 = 0$ in $\gamma_0$ it is clear that:

$$\eta_{\alpha} \in H_{\gamma_0}^1(S), \quad \bar{q} \in H_{\gamma_0}^1(S) \text{ et } \eta_3 \in H_{\gamma_0}^2(S), H_{\gamma_0}^2(S) = \{\eta_3 \in H^2(S), \eta_3 = \partial_{\vec{\nu}}\eta_3 = 0 \text{sur} \gamma_0\},$$

$\vec{\nu}$ is the external unit vector on the border of the medial surface.

The variational equation associated with the model problem reads as the following: find $(u(x), \bar{q}(x)) \in U_{ad}$ such that

$$E((u, \bar{q}); (v, \bar{y})) = L(v, \bar{y}) \text{ for all } (v, \bar{y}) \tag{23}$$

$E(.,.)$ is the coercive operator in the space of admissible displacement $U_{ad}$. Then $U_{ad}$ is a closed subset of $H^1(S) \times H^1(S) x H^2(S) \times H^1(S)$

where $U_{ad} = \left(H^1_{\Upsilon_0}(S) x H^1_{\Upsilon_0}(S) x H^2_{\Upsilon_0}(S) x H^1_{\Upsilon_0}(S)\right)$, $H^1_{\gamma_0}(S) = \{u \in H^1(S), u = 0 \text{ on } \gamma_0\}$, $H^m(S) = \{u \in L^2(S); \partial_\alpha u \in L^2(S), \alpha = (\alpha_1, \alpha_2), |\alpha| \le m\}$, and $L^2(S)$ are the set of square-integrable functions defined on the mid-surface $(S)$, and $\gamma_0$ is a portion of the boundary of the mid-surface where the displacement is fixed [39].

Let $A^{\alpha\beta\delta\tau} = \bar{\lambda} G^{\alpha\beta} G^{\delta\tau} + 2\bar{\mu} G^{\alpha\tau} G^{\tau\beta}$ be the shell characteristic component matrix for an isotropic linear elastic material with Lamé constants ($\bar{\lambda}$ and $\bar{\mu}$).

where $G^{\alpha\beta}$ is the contravariant component of the metric tensor of the shell.

Then, considerer the self-supporting roof of the building in the shape of an airplane $\Omega = S \times [-\frac{h}{2}, \frac{h}{2}]$ whose constitutive law is the following [41]:

$$\sigma^{ij}(u, q) = \bar{\lambda}\left(\in^\alpha_\alpha(u) + \in^l_l(q)\right) G^{ij} + 2\bar{\mu}\left(\in^{ij}(u) + \in^{ij}(q)\right) = C^{ijkl} \in_{kl}(u, q) \tag{24}$$

The solution $v$ can be decomposed as a sum of two displacements namely a plane strain displacement field $v(\eta) = (v_\alpha(\eta), \eta_3)$ and a stretching displacement $v(q) = (0, 0, q)$. Therefore the strain can also be decomposed as

$$\in_{ij}(v) = \in_{ij}(v(\eta)) + \in_{ij}(v(q)) = \in^\eta + \in^q \tag{25}$$

$$d\Omega = (G_1, G_2, A_3) dx^1 dx^2 dz = \left(1 - 2z\overline{H} + z^2\overline{K}\right) dz dS = \psi(x, z) dz dS$$

$$2\overline{H} = B^\alpha_\alpha \ , \ \overline{K} = det B,$$

where $B^\rho_\alpha = A^{\rho\gamma} B_{\gamma\alpha}$ and $B_{\gamma\alpha}$ denote the curvature tensor components and $A^{\rho\gamma}$ is the contravariant component of the metric of the mid-surface S. We have implicitly assumed that the characteristic parameter of the mid-surface $\chi = \frac{h}{R}$ is less than 1; h is half the thickness and R the minimum absolute value of its radius of curvature, with $v_\alpha = \eta_\alpha - z\left(\partial_\alpha \eta_3 + 2B^\rho_\alpha \eta_\rho\right) + z^2\left(B^\rho_\alpha B^\tau_\rho \eta_\tau + B^\rho_\alpha \partial_\rho \eta_3\right)$, $v_3 = \eta_3 + q(x, z)$ [41]

$$e_{\alpha\beta}(v) = (\nabla_\alpha v_\beta + \nabla_\beta v_\alpha - 2B_{\alpha\beta} v_3)/2. \tag{26}$$

$$\nabla_\alpha v_\beta = v_{\beta,\alpha} - \bar{\Gamma}^\rho_{\beta\alpha} v_\rho. \tag{27}$$

$$K_{\alpha\beta}(v) = \nabla_\alpha B^v_\beta \nu_v + B^v_\alpha \nabla_\beta \nu_v + B^v_\beta \nabla_\alpha \nu_v + \nabla_\alpha \nabla_\beta \nu_3 - B^\tau_\alpha B_{\tau\beta} \nu_3. \tag{28}$$

$$Q_{\alpha\beta}(\nu) = (B^v_\alpha \nabla_\beta B^\tau_v \nu_\tau + B^v_\alpha B^\tau_v \nabla_\beta \nu_\tau + B^v_\alpha \nabla_\beta \nabla_v \nu_3 + B^v_\beta \nabla_\alpha B^\tau_v \nu_\tau + B^v_\beta B^\tau_v \nabla_\alpha \nu_\tau + B^v_\beta \nabla_\alpha \nabla_v \nu_3)/2. \tag{29}$$

where $e$, $K$ and $Q$ (Gaussian tensor) are changes in the first, second and third fundamental forms of the mid-surface, respectively.

Therefore

$$
\begin{cases}
\in_{\alpha\beta}(v(\eta)) = e_{\alpha\beta}(\eta) - zk_{\alpha\beta}(\eta) + z^2 Q_{\alpha\beta}(\eta) = \in_{\alpha\beta}^{\eta}, \quad \in_{i3}(v(\eta)) = 0 = \in_{i3}^{\eta} \\
\in_{\alpha\beta}(v(q)) = q(x,z)\Upsilon_{\alpha\beta}(x,z) = \in_{\alpha\beta}^{q}, \\
\in_{\alpha3}(v(q)) = \partial_\alpha q/2 = \in_{\alpha3}^{q}, \quad \in_{33}(v(q)) = \partial_z q = \in_{33}^{q}
\end{cases}
\tag{30}
$$

The section warping tensor is given as follows.

$$
\gamma_{\alpha\beta}(x) = -\frac{1}{2}\left(\mu_\alpha^\rho B_{\rho\beta} + \mu_\beta^\rho B_{\rho\alpha}\right).
\tag{31}
$$

The Lamé and Poisson variable coefficients are expressed as follows:

$$
\xi = \xi(\omega, T, t)
$$

$$
\bar{v} = \bar{v}(\xi)
$$

$$
\bar{\lambda} = \frac{E(\xi, \varphi, \phi)\,\bar{v}}{(1 - 2\bar{v})(1 + \bar{v})};
\tag{32}
$$

$$
\bar{\mu} = \frac{E(\xi, \varphi, \phi)}{2(1 + \bar{v})}
\tag{33}
$$

where $\bar{v}$ and the Poisson's ratio depends on the variable and nonhomogeneous hydration coefficient, respectively.

- $\omega$ is the total water present in the concrete;
- $T$ is the temperature;
- t is time;
- $\varphi$ is thetotal creep coefficient.
- $\phi$ is the coefficient that takes into account ageing

The variational formulation of the boundary problem gives the following:

$$
E((u,\ \bar{q}); (v,\ \bar{y})) = \int_S \int_{-\frac{h}{2}}^{\frac{h}{2}} \sigma^{ij}(u,q) \in_{ij}(v, w\bar{y})\,\psi dz dS
$$

By integrating Equation (24) we have

$E((u,\ \bar{q}); (v,\ \bar{y}))$

$$
= \int_S \int_{-\frac{h}{2}}^{\frac{h}{2}} [A^{\alpha\beta\delta\tau}\epsilon_{\delta\tau}(u)\epsilon_{\alpha\beta}(v) + wA^{\alpha\beta\delta\tau}\Upsilon_{\delta\tau}\epsilon_{\alpha\beta}(u)\bar{y} + \bar{\lambda}w' G^{\alpha\beta}\epsilon_{\alpha\beta}(u)\bar{y} + \left(\bar{\lambda}G^{\alpha\beta}w' + wA^{\alpha\beta\delta\tau}\Upsilon_{\delta\tau}\right)\bar{q}\epsilon_{\alpha\beta}(v)]\psi dz dS
$$

$$
+ \int_S \int_{-\frac{h}{2}}^{\frac{h}{2}} \left[\left(w^2 A^{\alpha\beta\delta\tau}\Upsilon_{\delta\tau}\Upsilon_{\alpha\beta} + \bar{\lambda}ww' G^{\alpha\beta}\Upsilon_{\alpha\beta}\right)\bar{q}\,\bar{y} + \left(\bar{\lambda} + 2\bar{\mu}\right)(w')^2\bar{q}\,\bar{y} + \bar{\mu}\,G^{\alpha\beta}w^2\partial_\beta\bar{q}\partial_\alpha\bar{y}\right]\psi dz dS
$$

with $w' = \frac{dw}{dz}$

Let

$$D_{0w}^{\alpha\beta} = \int_{-\frac{h}{2}}^{\frac{h}{2}} \left[ wA^{\alpha\beta\delta\tau} \Upsilon_{\delta\tau} \right] \psi dz; \; D_{1w}^{\alpha\beta} = \int_{-\frac{h}{2}}^{\frac{h}{2}} z \left[ wA^{\alpha\beta\delta\tau} \Upsilon_{\delta\tau} \right] \psi dz; \; D_{2w}^{\alpha\beta} = \int_{-\frac{h}{2}}^{\frac{h}{2}} z^2 \left[ wA^{\alpha\beta\delta\tau} \Upsilon_{\delta\tau} \right] \psi dz$$

$$E_{0w}^{\alpha\beta} = \int_{-\frac{h}{2}}^{\frac{h}{2}} \left[ \bar{\lambda} \; G^{\alpha\beta} w' \right] \psi dz; \; E_{1w}^{\alpha\beta} = \int_{-\frac{h}{2}}^{\frac{h}{2}} z \left[ \bar{\lambda} \; G^{\alpha\beta} w' \right] \psi dz; \; E_{2w}^{\alpha\beta} = \int_{-\frac{h}{2}}^{\frac{h}{2}} z^2 \left[ \bar{\lambda} \; G^{\alpha\beta} w' \right] \psi dz$$

$$F_{0w}^{\alpha\beta}(\gamma) = \int_{-\frac{h}{2}}^{\frac{h}{2}} \left[ wA^{\alpha\beta\delta\tau} \Upsilon_{\delta\tau} + \bar{\lambda} G^{\alpha\beta} w' \right] \psi dz; \; F_{1w}^{\alpha\beta}(\gamma) = \int_{-\frac{h}{2}}^{\frac{h}{2}} z \left[ wA^{\alpha\beta\delta\tau} \Upsilon_{\delta\tau} + \bar{\lambda} G^{\alpha\beta} w' \right] \psi dz$$

$$F_{2w}^{\alpha\beta}(\gamma) = \int_{-\frac{h}{2}}^{\frac{h}{2}} z^2 \left[ wA^{\alpha\beta\delta\tau} \Upsilon_{\delta\tau} + \bar{\lambda} G^{\alpha\beta} w' \right] \psi dz; \; F_{w0}^{33}(\gamma) = \int_{-\frac{h}{2}}^{\frac{h}{2}} \left[ w^2 A^{\alpha\beta\delta\tau} \Upsilon_{\delta\tau} \Upsilon_{\alpha\beta} + \bar{\lambda} ww' G^{\alpha\beta} \right] \psi dz$$

$$F_{w1}^{33}(\gamma) = \int_{-\frac{h}{2}}^{\frac{h}{2}} \left[ \bar{\lambda} wG^{\alpha\beta} \Upsilon_{\alpha\beta} w' \right] \psi dz;$$

$$I_{ww}^{\alpha\beta}(\gamma) = \int_{-\frac{h}{2}}^{\frac{h}{2}} (\bar{\mu}) \; G^{\alpha\beta}(w)^2 \psi dz$$

Then an equivalent form of the variational equation is

$$E((u, \; \bar{q}); (v, \; \bar{y}))$$

$$= \int_S (N^{\alpha\beta}(u) e_{\alpha\beta}(v) + M^{\alpha\beta}(u) K_{\alpha\beta}(v) + M^{*\alpha\beta}(u) Q_{\alpha\beta}(v) \; dS + \int_S (D_{0w}^{\alpha\beta} e_{\alpha\beta}(u) - D_{1w}^{\alpha\beta} K_{\alpha\beta}(u) + D_{2w}^{\alpha\beta} Q_{\alpha\beta}(u)) \bar{y} dS$$

$$+ \int_S (E_{0w}^{\alpha\beta} e_{\alpha\beta}(u) - E_{1w}^{\alpha\beta} K_{\alpha\beta}(u) + E_{2w}^{\alpha\beta} Q_{\alpha\beta}(u)) \bar{y} dS + \bar{q} \int_S (F_{0w}^{\alpha\beta}(\gamma) e_{\alpha\beta}(v) - F_{1w}^{\alpha\beta}(\gamma) K_{\alpha\beta}(v) + F_{2w}^{\alpha\beta}(\gamma) Q_{\alpha\beta}(v)) dS$$

$$+ \int_S \bar{q} \; \bar{y}(F_{w0}^{33}(\gamma) - F_{w1}^{33}(\gamma) + F_{w2}^{33}(\gamma)) dS + \int_S \left( I_{ww}^{\alpha\beta} \partial_\beta \bar{q} \partial_\alpha \bar{y} \right) dS \; .$$

where setting $\psi \approx 1, G^{\alpha\beta} \approx A^{\alpha\beta}$ yields

$$L(v, \; \bar{y}) = \int_S (\bar{P}^i v_i + \bar{P}^4 \bar{y}) dS + \int_{\gamma_1} (q^i v_i + q^4 \bar{y}) d\gamma + \int_{\gamma_1} (m^\alpha \theta_\alpha(v) \, d\gamma$$

Here $(\bar{P}^i, \bar{P}^4)$, $(q^i, q^4)$, $m^\alpha$ and $\theta_\alpha$ are, the resultant surface force, shear force, moment density, and opposite angle of rotation of a section at the border, respectively.

where $\bar{P}^4 = \int_{-\frac{h}{2}}^{\frac{h}{2}} f^3 w dz + w\left(\frac{h}{2}\right) \bar{p}_+^3 - w\left(-\frac{h}{2}\right) \bar{p}_-^3$ and $\theta_\alpha(v) = -(\partial_\alpha v_3 + B_\alpha^\rho v_\rho)$ is the virtual angle of rotation of a cross section at the free border of the mid-surface in the direction of the base vector $A_\alpha$.

$$N^{\alpha\beta} = \frac{Eh}{1-\bar{v}^2} \left[ (1-\bar{v}) e^{\alpha\beta}(u) + \frac{\bar{v}(1-\bar{v})}{1-2\bar{v}} e_\rho^\rho(u) A^{\alpha\beta} \right] + \frac{Eh^3}{12(1-\bar{v}^2)} \left[ (1-\bar{v}) Q^{\alpha\beta}(u) + \frac{\bar{v}(1-\bar{v})}{1-2\bar{v}} Q_\rho^\rho(u) A^{\alpha\beta} \right] \tag{34}$$

$$M^{\alpha\beta} = \frac{Eh^3}{12\left(1-\bar{v}^2\right)} \left[ (1-\bar{v}) K^{\alpha\beta}(u) + \frac{\bar{v}(1-\bar{v})}{1-2\bar{v}} K_\rho^\rho(u) A^{\alpha\beta} \right] + \frac{Eh^3}{12\left(1-\bar{v}^2\right)} \left[ (1-\bar{v}) e^{\alpha\beta}(u) + \frac{\bar{v}(1-\bar{v})}{1-2\bar{v}} e_\rho^\rho(u) A^{\alpha\beta} \right] \tag{35}$$

$$M^{*\alpha\beta} = \frac{Eh^5}{80\left(1-\bar{v}^2\right)} \left[ (1-\bar{v}) Q^{\alpha\beta}(u) + \frac{\bar{v}(1-\bar{v})}{1-2\bar{v}} Q_\rho^\rho(u) A^{\alpha\beta} \right] \tag{36}$$

We use isoparametric Lagrangean symplectic type triangle finite elements which are based on complete polynomial bases. The finite element used has 4 (four) degrees of freedom at the vertex nodes (3 displacements and 1 stretching) and 1 (one) degree of freedom at the midpoints of edges (1 displacement w). The finite element used is of the (MT6) family described in the works of Djopkop et al [40] and [41]. In Paper [41], three tests (hemisphere under diametrically opposed load, pinched cylinder, and sphere under uniform pressure) are carried out and the convergence results are discussed with the finite element method presented.

## 4. Numerical implementation

In this section, after presenting the part of the self-supporting roof to be applied, we propose the coupling technique adopted to integrate all delay effects into the behavior law.

### 4.1. Shell structure geometry

The modelled concrete shell is part (B) of the self-supporting roof of an aircraft-shaped building (Fig 1).

The input data in the algorithm are $inc = increments\_of\_time$, $tps\_sim = time\_of\_simulation$.

The following steps promote understanding of the nonlinear algorithm called Djopkop's algorihm, where a material of rheological nonlinearity is established (Fig 2).

a) *Temporal and spatial monitoring of hydration $\alpha$;*

b) *Evaluation of the mechanical parameters of the concrete $(\bar{\lambda}, \bar{\mu}, \bar{\nu})$;*

c) *Consideration of concrete shrinkage; consider $t_0$ as the initial temp after application of the mechanical load;*

d) *Determination of the elastic strain $\in^e$ using N theory;*

e) *Superposition of $\in^e$ to the shrinkage strain $\in^{sh}$;*

f) *Calculation of the new shell thickness $h_1(x, t_o) = h + \int_{-h/2}^{h/2} \epsilon_{33} dz$;*

g) *Injection of $h_1(x, t_o)$ in the N model and determination of the stress state of the structure.*

   ***Moment $t_i$ $i > 0$***

h) *Calculation of the damage parameter;*

i) *Consideration of ageing and creep (which leads to changes in mechanical parameters);*

j) *Integration into the model (which leads to obtaining the strain integrating creep $\epsilon^{ec}$);*

k) *Superposition $\epsilon = \epsilon^{ec} + \epsilon^r(t_i) \rightarrow h_{j+1}(x, t_i)$;*

l) *Resumption from h) until the end of the loop (simulation).*

Note: We assume that 28 days is as the time of application of the mechanical load. At this time, we assume that the structure has already been erected, so we can calculate the creep.

The calculation of the new thickness of the shell is performed via the following relation:

$$\widetilde{h}(x, t) = h + \int_{-h/2}^{h/2} \epsilon_{33} dz$$

(37)

### 4.2. Main data

In this work, we cast prismatic specimens with dimensions of 7 cm × 7 cm × 28 cm, in the laboratory environment, (temperature 22°C ± 1°C, relative humidity 70 ± 5%) similar to the in situ conditions of the structure. The goal is to monitor the drying parameters.

**4.2.1. Concrete mixture.** The cement used in this study is ordinary Portland cement CEM I 42.5 R. The uniaxial compressive strength of the concrete was determined on a cubic sample on the of 10 cm side at 28 days. The mixture proportions and the average compressive strength ($f_c$) are presented below (Table 1):

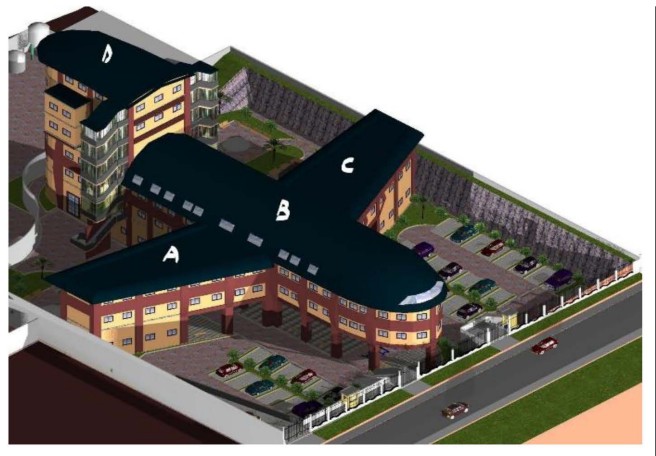

| | |
|---|---|
| *Fig 1.a: 3D view in pictures* | |

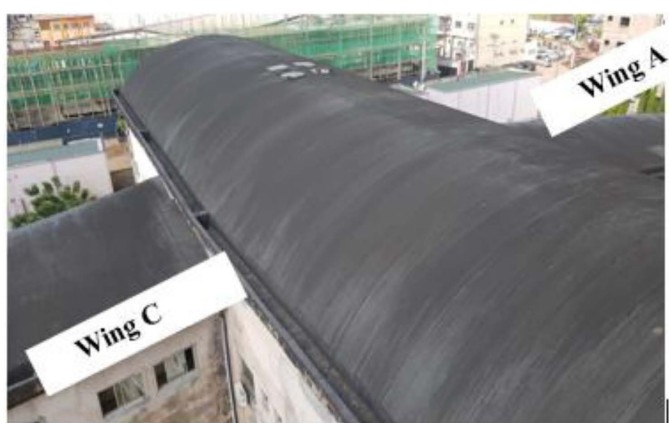

| | |
|---|---|
| | *Fig 1.b: Real 3D view* |

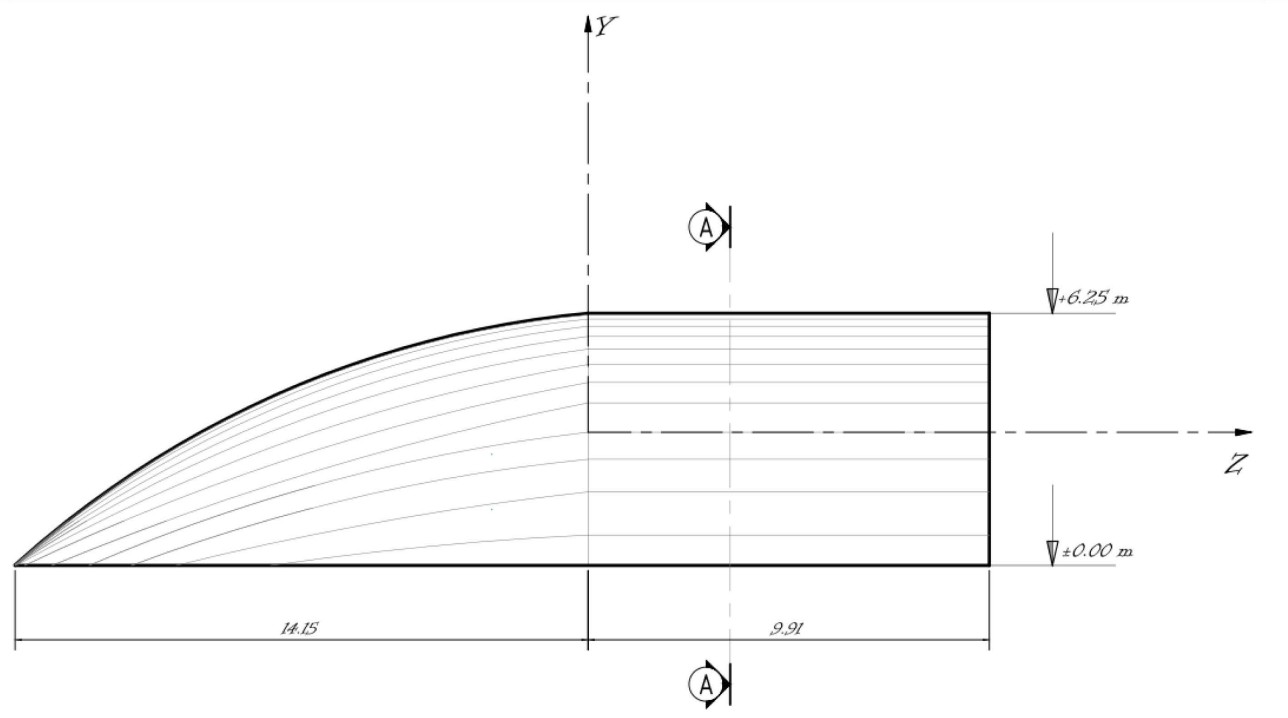

*Fig 1.c: Left fronts of part B*

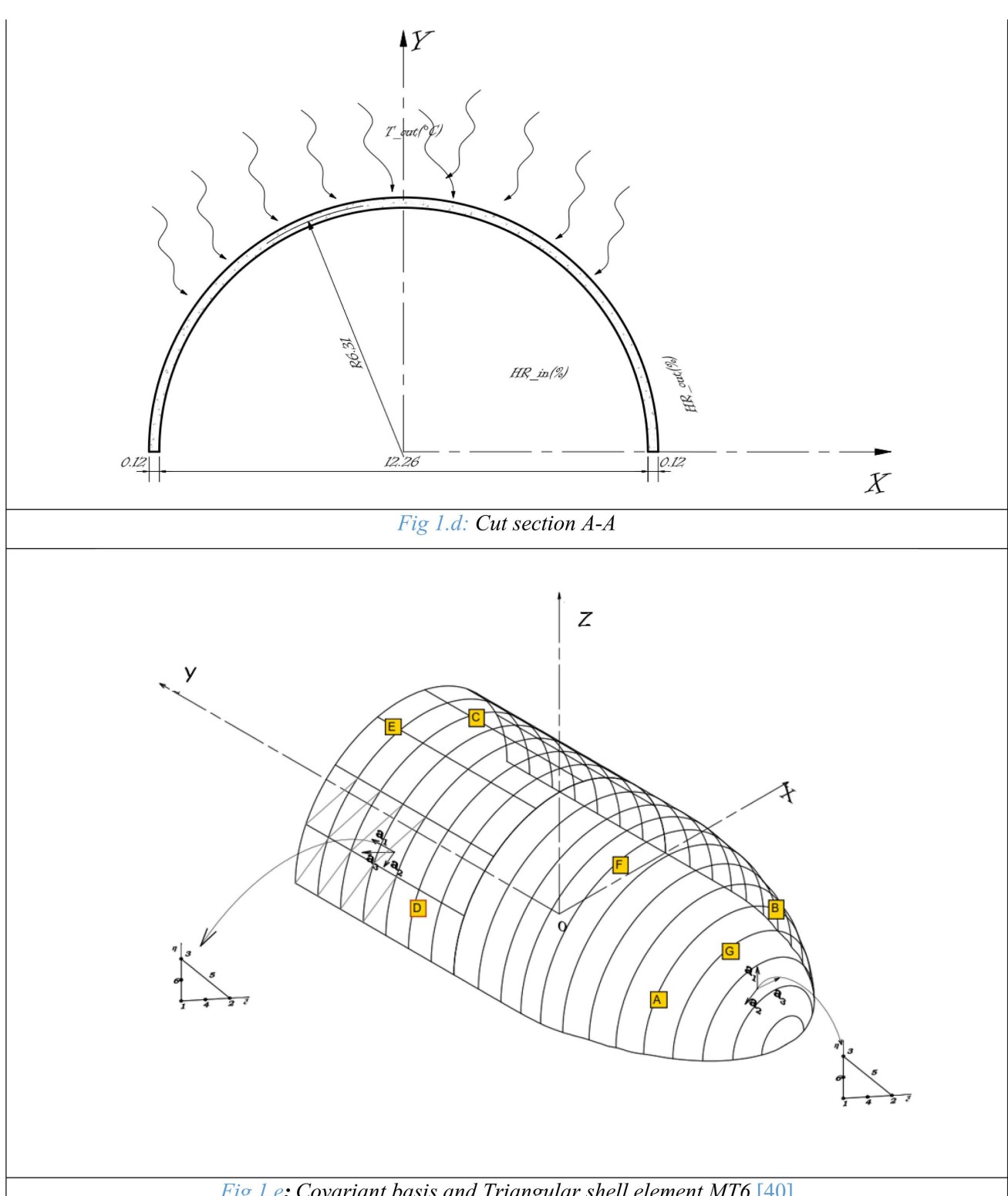

*Fig 1.d: Cut section A-A*

*Fig 1.e: Covariant basis and Triangular shell element MT6* [40]

**Fig 1. Shell structure geometry.** (a) 3D view in pictures. (b) Real 3D view. (c) Left fronts of part B. (d) Cut section A-A.(e) Covariant basis and Triangular shell element MT6 [40].

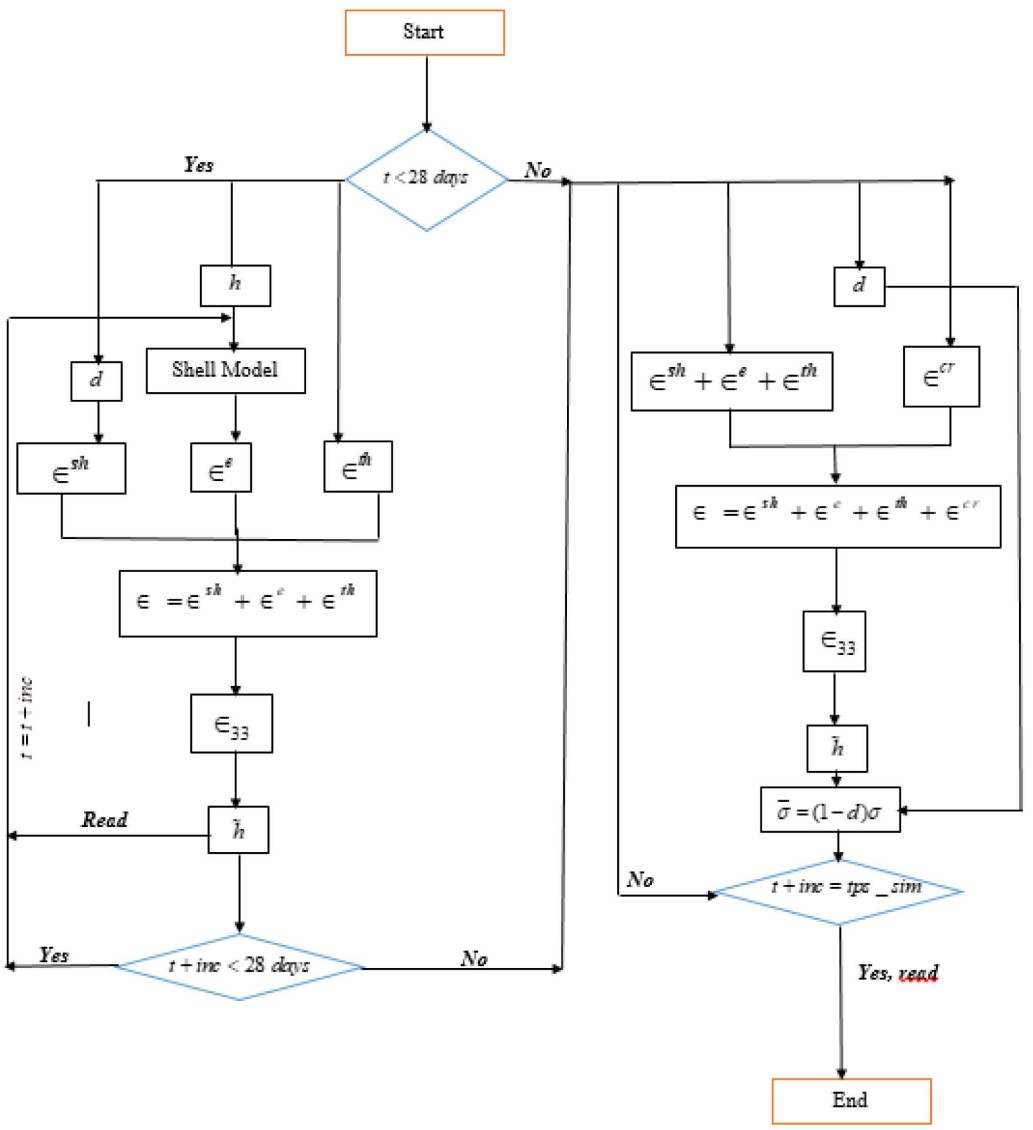

$\in^e$ : elastic strain tensor;

$\in^{sh}$ : shrinkage strain tensor;

$\in^{th}$ : thermal strain tensor;

$\in^{cr}$ : creep strain tensor;

$\in$ : total strain tensor;

$d$ : damage coefficient;

**Fig 2. Calculation algorithm implementing the proposed model.**

**4.2.2. Mass loss, shrinkage measurement and crush test.** The same process described in [22] was used for mass loss (see Fig 3a, Fig 3b, Fig 3c, Fig 3d and Fig 4) and shrinkage measurements. Concrete samples of size $7cm \times 7cm \times 28cm$ and other prismatic samples of size $10cm \times 10cm \times 10cm$ were cast. The shrinkage measurements started 24 hours later in the controlled environment immediately after the removal of the formwork. Autogenous shrinkage

**Table 1. Composition of concrete.**

| Composition of concrete | | | | | |
|---|---|---|---|---|---|
| Water (W) | Cement (c) | Sand (s) | Gravel 0/15 | W/c | $f_c$ |
| 175 L | 350 kg | 750 kg | 1150 kg | 0.5 | 25MPa |

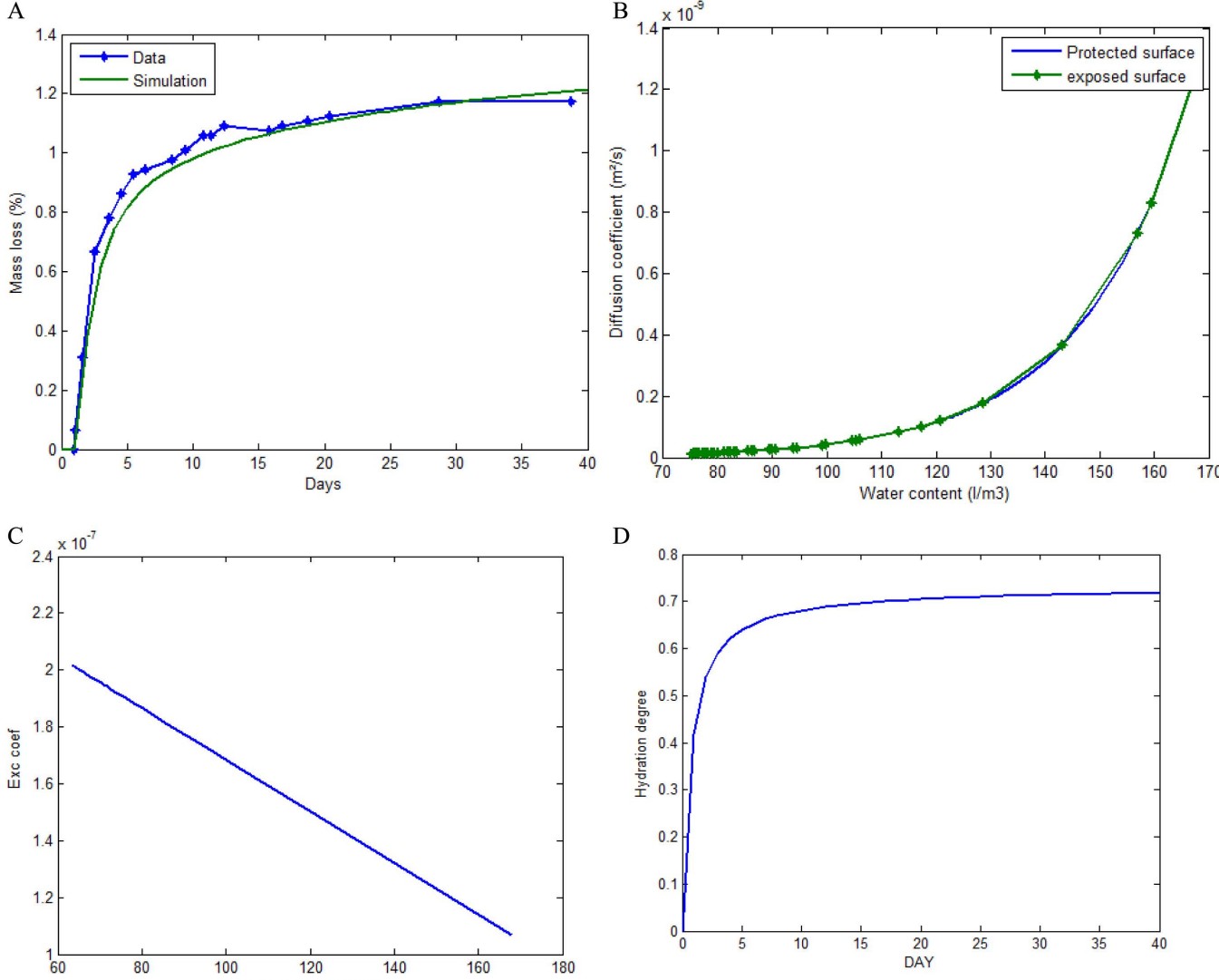

**Fig 3. Calculation algorithm implementing the proposed model.** (a). Mass loss. (b) Diffusion coefficient. (c) Exchange coefficient. (d) Hydration degree.

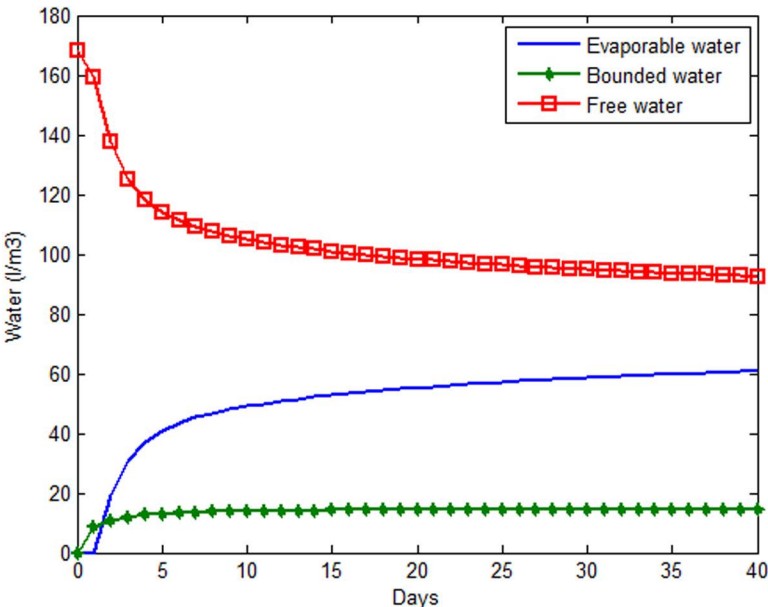

**Fig 4. Average water variation in the mixture.**

A B

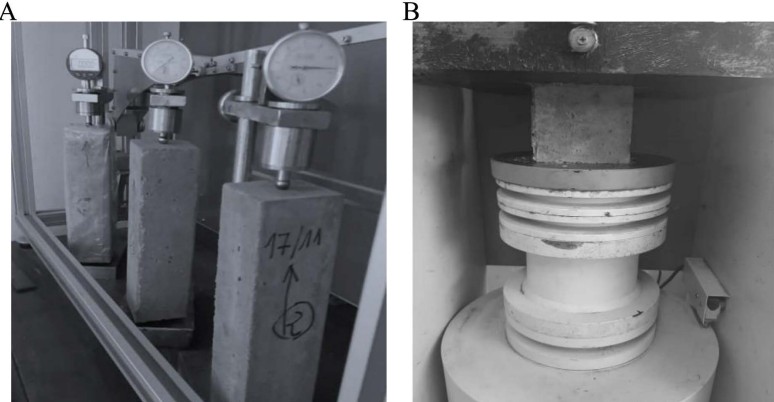

**Fig 5. Prismatic samples.** (a). Measurement of endogenous shrinkage and drying shrinkage. (b) crush test.

was measured on 3 samples of size $7cm \times 7cm \times 28cm$ sealed with an adiabatic waterproof sheet to prevent drying. In addition, three $10cm \times 10cm \times 10cm$ prismatic samples, with five faces sealed with painted waterproof sheets were used for monitoring mass loss due to drying.

The last three prismatic samples were used for crushing tests to confirm the compressive strength ($f_c$) (see Fig 5).

## 5. Results and discussion

An experimental campaign was conducted using the properties of the constitutive materials of the self-supporting concrete shell roof. These data were mainly used to simulate total shrinkage. With these results, we next proceeded by implementing our (thermochemical and hydromechanical (TCHM)) approach to monitor strain evolution in the shell.

 

## 5.1. Intermediate results

To obtain the strain evolution in the shell, intermediate results are processed via the algorithm presented in Section 4. The results obtained include the drying parameters, water evolution, shrinkage and thermal strain. These results make it possible to follow the evolution of the mechanical properties presented in Section 3.5.

i. Drying parameters.

The model curve and experimental curve are presented. In Fig 3a, a model curve reproduced the experimental curve in blue. Therefore, the model curve is used to evaluate the material properties $\bar{A} = 5.186 \times 10^{-12}$ and $\delta = 9.06 \times 10^{-10}$ (parameters presented in Equation (3) and (4)). Fig 3b, presents a model coefficient of diffusion of the exposed face and the unexposed face that depends on water content obtained from the experimental data. Fig 3c, presents a model coefficient of exchange that depends on the water content and is obtained from experimental data. The model time dependent coefficient of hydration, obtained from the experimental data is presented in Fig 3d.

All the calibration parameters are obtained by fitting the experimental results. The rest of the results are then derived from the model.

ii. Time evolution of the water content.

The water content at all times is decomposed into free water, bound water and evaporated water. The numerical calculation carried out on a section provides the values at each of the points in the domain. An average is subsequently made of the values obtained in an increment of time steps. The free water drops rapidly during the first 5 (five) days and then evolves asymptotically to a fixed value. This loss of free water is due to hydration and evaporation as shown in Fig 4. The hydration coefficient which reflects the hydration phenomenon, therefore evolves over time and consequently promotes the development of Young's modulus (Section 3.1.1). This mechanical parameter therefore depends indirectly on time and position.

iii. Time evolution of shrinkage.

Fig 6.a and Fig 6.b present the evolution of the endogenous and drying shrinkage strains. A linear interpolation is performed to approach a mathematical model to follow the strain in time. One can deduce the values of $k = 286.72 \times 10^{-6}$ and $p = 294.05 \times 10^{-6}$ (Formula (6) and (7)) with correlation coefficients 0.9837 and 0.9558, respectively.

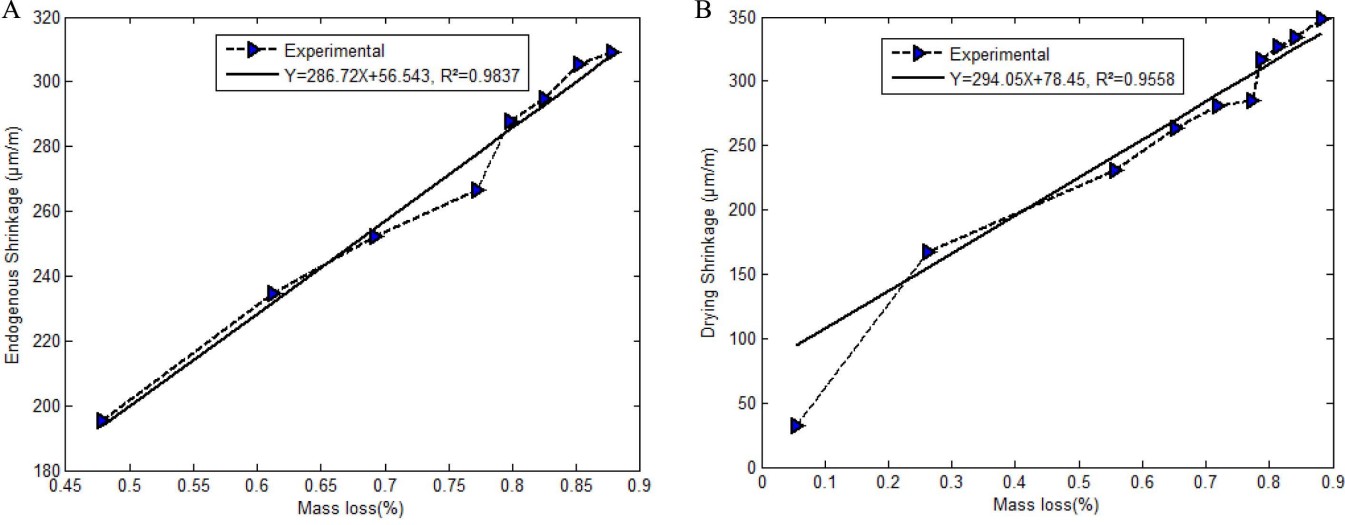

**Fig 6. Fitting of coefficients.** (a).Fitting of the coefficient k. (b) Fitting of the coefficient p.

The endogenous shrinkage increases less than the total shrinkage does and has the same direction. Finally, from the 12th day after the removal of the formwork, all the variations are small. We observed that the endogenous shrinkage curve shows a quasilinear evolution limited by $320 \mu m/m$. The total shrinkage curve tends asymptotically towards $800 \mu m/m$ after forty days (Fig 7). This finding is in agreement with the results shows in Fig 4. The total shrinkage is proportional to the evolution of the evaporated water. However, the latter tends asymptotically towards the limiting value.

**5.1.4 Time evolution thermal strain.** First, we present the evolution of the temperature in the year 2020 [45] of the city of Douala, in the Littoral region, of Cameroon (Fig 8). The structure is located in this city.

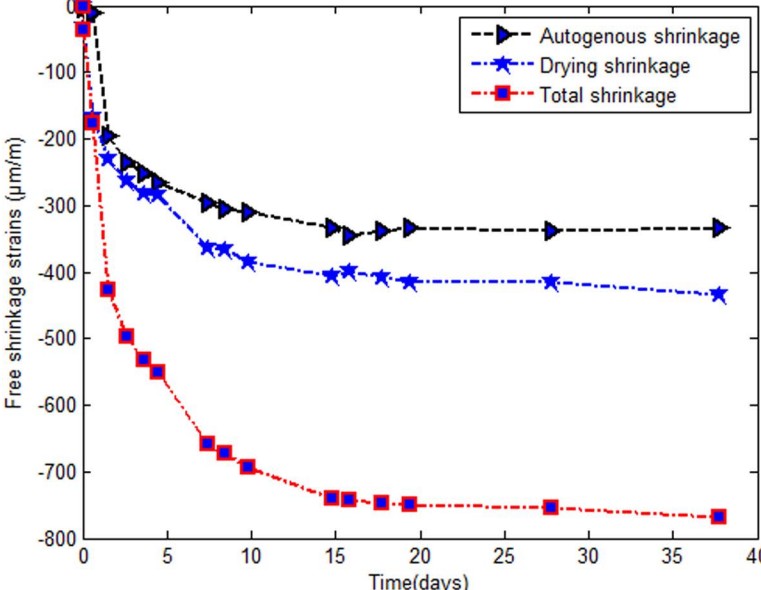

**Fig 7. Simulation of shrinkage in time.**

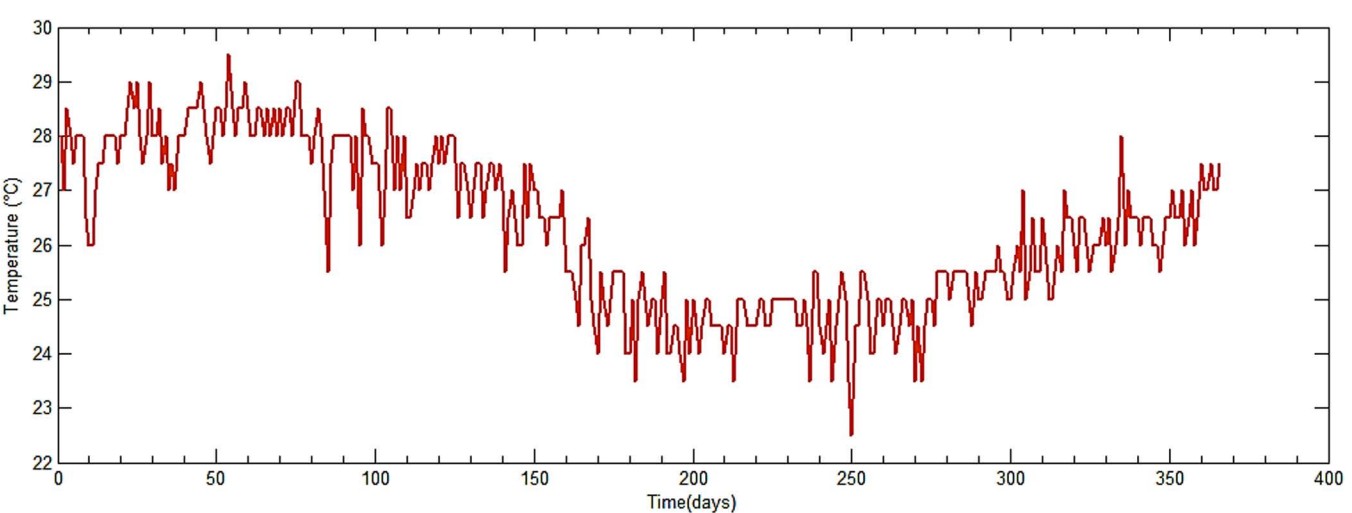

**Fig 8. Average temperature over the year 2020.**

The curve plotted is the average temperature over time in days. We use Equation (8) to deduce the evolution of the thermal shrinkage shown in Fig 9. We used the 50-day average temperatures.

The delay effects considered in this paper are shrinkage and creep. Creep is integrated into the model using Equation (14). Shrinkage is taken into account in previous sections of this work (see Sections 5.1.3 iii and 5.1.4)

### 5.2. Results for the self-supporting roof

The time evolution deflection, thickness variation and strain through the thickness are presented for seven (7) points (3 on the spherical section, 1 at the junction between the cylindrical and spherical sections and 3 on the cylindrical section), as show in Fig 10. The coordinates and their corresponding nodes are indicated in Table 2. We use a time step of 5 days. For an explicit calculation, the calculation code was performed in MATLAB.

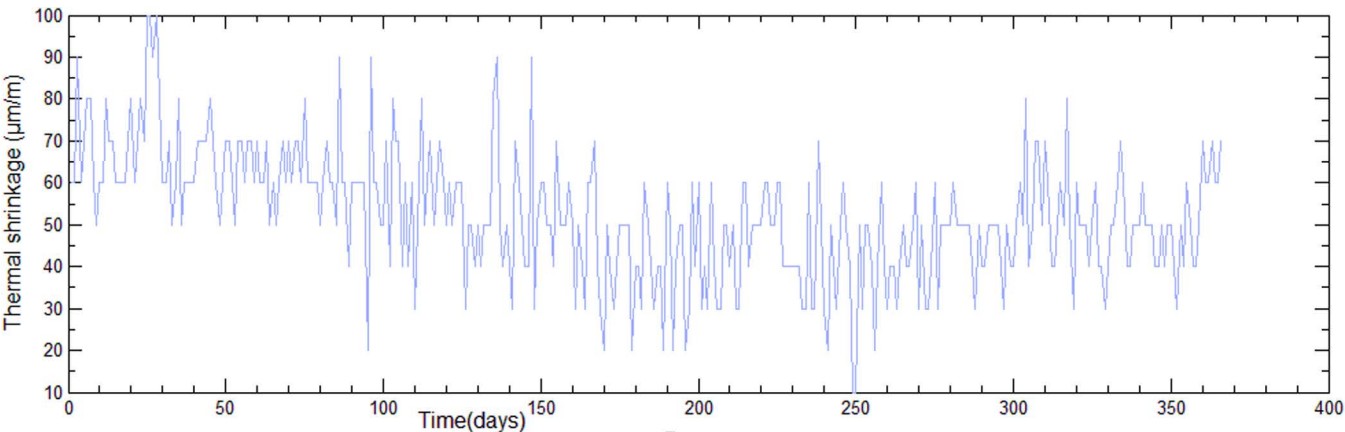

**Fig 9. Time evolution of thermal shrinkage.**

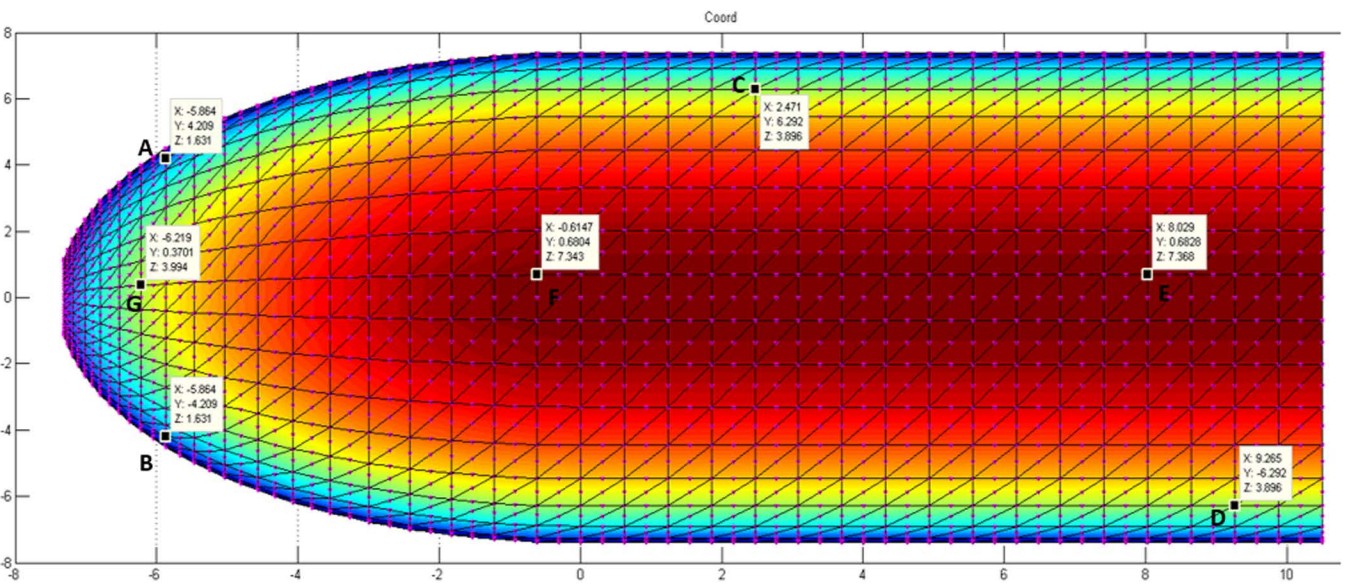

**Fig 10. Position and coordinates of the different points.**

**5.2.1. Damage evolution.** The calibration parameters for the simulation of the damage variable using the revisited MAZARS model are presented here($A_t = 0.85$; $B_t = 10000$; $A_c = 1.5$; $B_c = 1200$; $\varepsilon_{do} = 0.75 * 10^{-4}$; $j = 0.7$).

The calculation of the damage factor with the model used involves the main strains and stresses. The strain and stress tensors through the thickness are calculated for a finite number of points. Fig 11 shows the damage to the top sheet (extrados). This choice is justified by the fact that extrados are the areas most exposed to desiccation. The trend is the same for all the different nodes considered. During the first five days the damage factor is approximately 0 (zero) for all the nodes. From the sixth day, it gradually increases from 0 (zero) to values between 0.25 and 0.31 and increases

**Table 2. Coordinates and positions of the selected points.**

| Points | X (m) | Y(m) | Z(m) | Positions |
|---|---|---|---|---|
| A | −5.8639 | 4.2091 | 1.6306 | 111 |
| B | −5.8639 | −4.2091 | 1.6306 | 124 |
| C | 2.4706 | 6.2916 | 3.8956 | 382 |
| D | 9.2647 | −6.2916 | 3.8956 | 591 |
| E | 8.0290 | 0.6828 | 7.3680 | 549 |
| F | −0.6147 | 0.6804 | 7.3430 | 297 |
| G | −6.2190 | 0.3701 | 3.9940 | 99 |

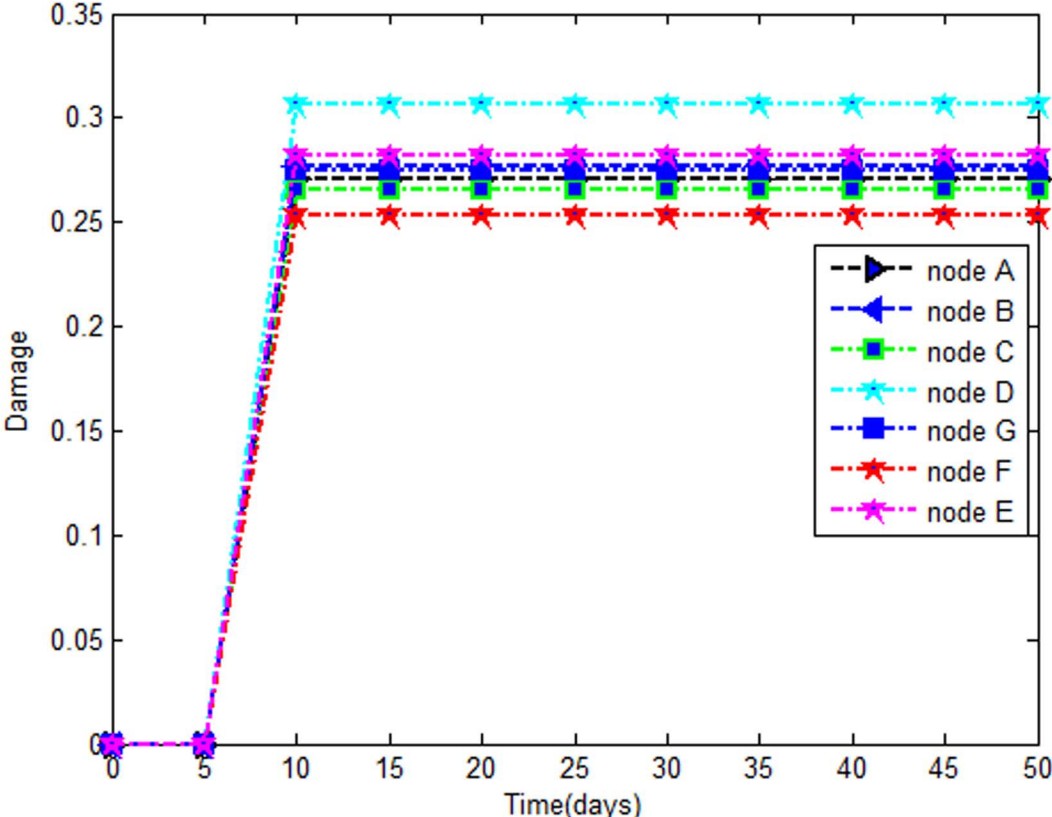

**Fig 11. Time evolution of the damage.**

very slowly around these values. The calculated damage comprises contributions of the delay effect phenomenon and mechanical loading.

**5.2.2. Evolution of the deflection.** Fig 12 shows that the absolute value of the deflection is time dependent. The points on the cockpit deflect less than those on the cylindrical part. Globally the defection on the cylindrical section is almost three times greater. The double curvature of the spherical part offers more rigidity as shown in the shell equation [39] Table 3.

In addition a sensitivity study is presented in Tab 3. Overall the displacements decrease when the $h/R$ (khi) ratio increases between 0 and 20 days, and he ratio between the displacements corresponding to $h/R = 0.038$ and $0.0192$

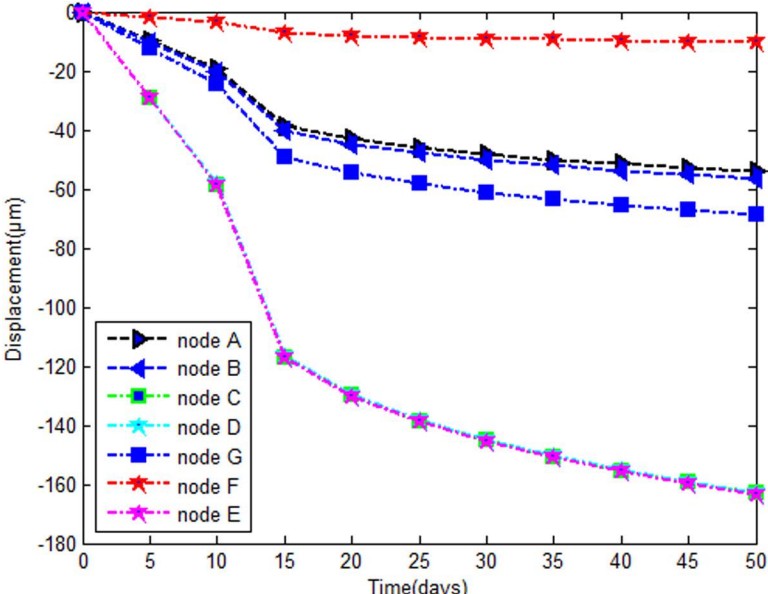

**Fig 12. Time evolution of the deflection.**

**Table 3. Deflection of node E of the shell according to the ratio $h/R$, h is the thickness, R is the radius of curvature (6,25m) (see Fig 1.c).**

| $t$ (day) | Ratio $h/R$ | |
|---|---|---|
| | 0.0192 | 0.038 |
| | Displacement ($\mu$m) | Displacement ($\mu$m) |
| 0 | 0 | 0,0 |
| 5 | 25,2 | 11,5 |
| 10 | 60,01 | 27,3 |
| 15 | 115,12 | 52,3 |
| 20 | 130,23 | 59,2 |
| 25 | 138,41 | 62,1 |
| 30 | 140,09 | 62,8 |
| 35 | 150,48 | 67,5 |
| 40 | 155,09 | 69,5 |
| 45 | 158,81 | 70,9 |
| 50 | 165,27 | 73,8 |

is 2.2. Then, between 25 and 40 days, we obtain 2.23 and finally 2.24 at the end. We can see that the ratio oscillates between 2.22.

**5.2.3. Thickness variation.** The time evolution of the thickness in both parts (spherical and cylindrical) is quasilinear and identical (Fig 13) during the simulation interval, $10\mu m$ at the 10th day and $85\mu m$ at the 50th day. In a static elastic analysis [40] of a thick ring with an initial thickness of 12 cm, the thickness variation obtained is $4.5\mu m$ under a uniform radial load of 6 bars. For the 6.8 cm thin ring, under a radial load of 5.49 bars, the thickness variation obtained is $10\mu m$. In this nonlinear analysis, only the self-weight (0.03 bar) is considered a mechanical load.

Fig 13 shows that for a thin shell, after 10 days, under a mechanical loading of 0.03 bar coupled with delay effects, the thickness variation on the 10th day is already $10\mu m$, unlike in the static elastic analysis above where the mechanical loading was 5.49 bars which is equal to 183*0.03 bars. This finding shows that delay effects have a greater contribution to the evolution of thickness variation. The advanced model presented herein captures a more precise thickness variation, and consequently, better stress-strain evaluations for a more accurate design.

**5.2.4. Total strain evolution.** In Fig 14, from the superposition of the different curves the following can be deduced:

i)   The deformation is greater in the cylindrical part than in the spherical part (cockpit) this is in line with the kinematics proposed in Equation (26), since the energy contribution of the Gauss tensor ($Q_{\alpha\beta}$) is greater in a spherical shell than in a cylindrical shell.

ii)  Before the 15th day, the various curves do not follow a preferred direction. Nevertheless, we note an increase in deformation on these days. This variation is attributed to hydration kinetics, which are accentuated in the first few days, shrinkage phenomena and creep. From the 15th day onwards, the curves evolved to individually accepted asymptotic values, demonstrating the predominance of creep deformation under self-weight on the one hand, and hydration kinetics with its induced effects on the other hand.

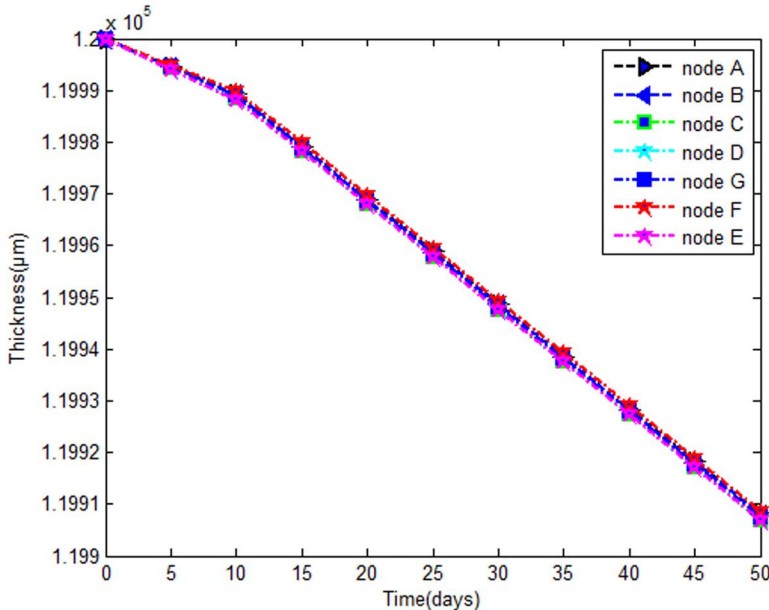

**Fig 13. Evolution of the thickness in the time.**

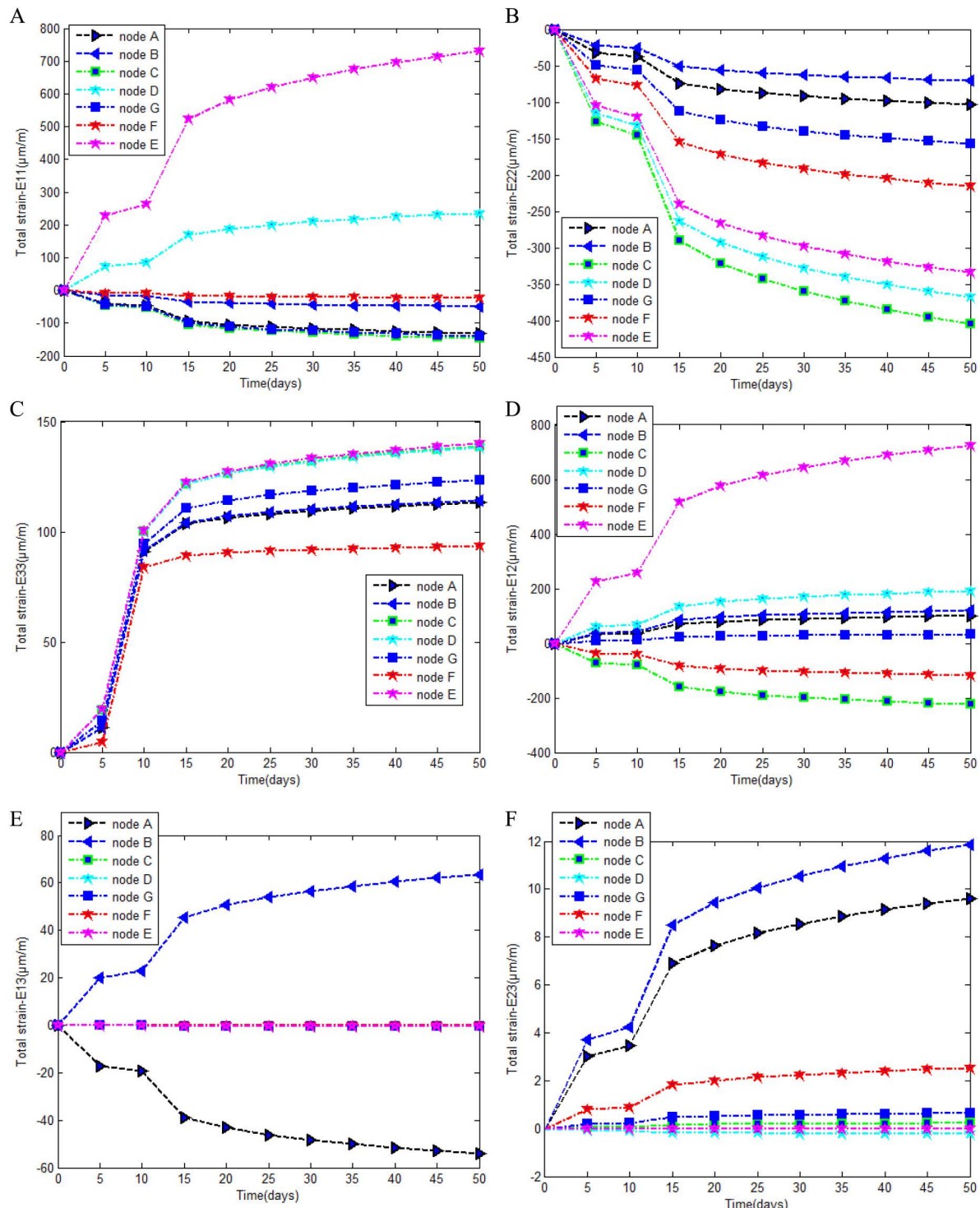

**Fig 14. Strain evolution.** (a).Evolution of the strain ($\in_{11}$) in the time. *(b).Evolution of the strain ($\in_{22}$) in the time.* (c) Evolution of the strain ($\in_{33}$) in the time.(d) Evolution of the strain ($\in_{12}$) in the time.(e) Evolution of the strain ($\in_{13}$) in the time. (f) Evolution of the strain ($\in_{23}$) in the time.

## 6. Conclusion

In this work, we present a new method based on the TCHM approach, that is capable of integrating the time dependent delayed deformations of concrete shell structures due to coupled phenomena such as hydration, shrinkage, and creep. A Logigram for the different calculation strategies is established. The software parameters were adjusted through experimental data. The purpose of this study is to arrive at a behaviour law for the concrete material and a law that integrates the specificities of this material. Existing standards, such as the Eurocode, propose a method for considering external dissipation by means of the coating. Controlling the consideration of internal dissipation by means of a behaviour law would be beneficial. This work follows this approach; we have seen the impact of the results between a calculation in linear elasticity and a model in a dissipative environment. The implemented model offers a time prediction depending on the calculation capacity of the computer. This numerical and experimental study has raised new questions about the consequences of drying on the durability of reinforced concrete shells. These questions can lead to new research perspectives: evaluating the influence of the level of damage to concrete shells, analysing the state of stress in the structure when it is subjected to long-term exposure, and considering certain pathologies in the assessment of the state of stress in the thickness of the shell.

## Supporting information

**S1 File. COCKPIT.**
(RAR)

## Author contributions

**Conceptualization:** Landry Djopkop Kouanang, Merlin Bodol Momha.

**Data curation:** Landry Djopkop Kouanang, Jean Chills Amba.

**Formal analysis:** Landry Djopkop Kouanang, Daniel Ambassa Zoa, Jean Chills AMBA, Joseph Nkongho Anyi, Robert Nzengwa.

**Investigation:** Landry Djopkop Kouanang, Merlin Bodol Momha, Daniel Ambassa Zoa.

**Methodology:** Landry Djopkop Kouanang, Jean Chills AMBA, Robert Nzengwa.

**Project administration:** Merlin Bodol Momha.

**Resources:** Landry Djopkop Kouanang.

**Software:** Landry Djopkop Kouanang, Merlin Bodol Momha, Daniel Ambassa Zoa.

**Supervision:** Jean Chills Amba.

**Validation:** Merlin Bodol Momha, Robert Nzengwa.

**Visualization:** Daniel Ambassa Zoa, Joseph Nkongho Anyi.

**Writing – original draft:** Landry Djopkop Kouanang.

**Writing – review & editing:** Landry Djopkop Kouanang, Robert Nzengwa.

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
