## [Decision Letter · Decision Letter 0]

PONE-D-24-21940ANALYSIS OF A SELF-SUPPORTING SHELL CONCRETE ROOF WITH NONLINEAR COUPLED EVOLUTIVE MATERIAL PARAMETERS.PLOS ONE

Dear Dr. DJOPKOP,

Thank you for submitting your manuscript to PLOS ONE. After careful consideration, we feel that it has merit but does not fully meet PLOS ONE’s publication criteria as it currently stands. Therefore, we invite you to submit a revised version of the manuscript that addresses the points raised during the review process.

The manuscript has been evaluated by three reviewers, and their comments are available below.

The reviewers have raised a number of concerns that need attention. They request improvements to the clarity of the methods and writing. Additional statistical analysis is also required.

Could you please revise the manuscript to carefully address the concerns raised?

We look forward to receiving your revised manuscript.

Kind regards,

Helen Howard

Staff Editor

PLOS ONE

Journal Requirements:

5. Please amend the manuscript submission data (via Edit Submission) to include author “Merlin Bodol Momha,  Jean Chills Amba,  Ambassa Zoa, Joseph Nkongho Anyi, Robert Nzengwa”

Reviewers' comments:

Reviewer's Responses to Questions

**Comments to the Author**

1. Is the manuscript technically sound, and do the data support the conclusions?

Reviewer #1: Partly

Reviewer #2: Yes

Reviewer #3: Partly

2. Has the statistical analysis been performed appropriately and rigorously? 

Reviewer #1: Yes

Reviewer #2: No

Reviewer #3: N/A

3. Have the authors made all data underlying the findings in their manuscript fully available?

Reviewer #1: No

Reviewer #2: Yes

Reviewer #3: No

4. Is the manuscript presented in an intelligible fashion and written in standard English?

Reviewer #1: Yes

Reviewer #2: Yes

Reviewer #3: No

5. Review Comments to the Author

Reviewer #1: • Overall Strengths: The manuscript offers a novel contribution to the field by integrating multiple phenomena that affect concrete performance over time, making it particularly valuable for long-term structural design applications. The proposed TCHM model is robust and well-supported by experimental data.

• Areas for Improvement:

1. Data Interpretation: Provide more detailed explanations for intermediate results, particularly regarding the effects of hydration and delayed deformations on the overall structural behavior.

2. Statistical Analysis: Consider performing sensitivity analysis to further validate your model and provide deeper insights into the impact of key parameters.

3. Data Sharing: Ensure that all underlying data, including raw measurements, are made fully available in accordance with PLOS ONE's data policy.

4. Language: Simplify the presentation of some technical content to improve readability and correct minor typographical and grammatical errors.

Conclusion: The manuscript is scientifically sound and offers important insights for concrete shell design. With minor revisions to improve data accessibility, statistical rigor, and clarity, it is suitable for publication.

Reviewer #2: The paper covers an important topic, but the authors need to strengthen the logical reasoning and clarity of the results. Specifically:

1. Revise the abstract to include details about the experimental work, outcomes, and recommendations.

2. Provide more in-depth explanations in the section discussing linear shrinkage.

3. Add a paragraph outlining potential future research directions based on the current study.

4. Include a paragraph at the end of the introduction emphasizing the research significance of the study.

5. Update the references with more recent research, as there are no citations from 2024 and only two from 2023, to better support the ideas and highlight the novelty of the work.

Reviewer #3: This article aims to describe a more complete framework by which to evaluate structural deformation in thin-shelled concrete dome structures: by including the contributions of thermochemical and hydromechanical behavior (e.g. creep) of concrete, in addition to typical elastic deformation. The paper then showcases the results of the proposed methodology by an analysis of a structure designed like an aircraft, particularly in the fuselage-like portion.

While the proposed article has potential, and appears to investigate an area of interest, the article lacks clarity in a number of areas. This makes the article difficult to evaluate from the perspective of novelty, completeness, and value of the proposed method. Particularly, the article needs to better address the following areas:

1. Many of the terms that are used are not defined, and not motivated. Particularly,

a. It is not clear why the Kronecker delta is only define don the midsurface.

b. What is \mu^\rho_\alpha, which is on the last line of the first page of the document? What does it physically correspond to?

c. Does the document use summation notation on indices or not?

d. What is the physical meaning of \psi (the first equation of the second page?

e. When \omega_L is introduced on page 5, it should be explicitly mentioned (using the language from the start of the text) that this is the free water content.

f. A description of the variables and parameters in Equations 2 and 3 are not present in the text. What are these? Without physical intuition and meaning, it is hard to motivate why the method of choice by the authors is appropriate.

g. What does “The drying parameter A and \delta are numerically adjusted after the resolution” mean (see line 141 of the text)? Are these parameters that are fit based on experimental data? Or to match a given deformation? Please clarify.

h. Is k of line 148 the same as \kappa in line 147? What do these calibration parameters correspond to? What is their physical meaning?

i. Variables in Section 3.3 lack definition, explanation, and intuition. Please label these variables and describe what they mean. What is the intuition that a practitioner can use to understand these? Also, for equation 13, what is the meaning of the numbers 1000 for concretes with silica fume and 3200 for those without silica fume? What is the repercussion for changing these numbers in this way? Where did these numbers come from?

j. Should Ac and Bc of equations 179 and 180 be A_c and B_c, where the _ denotes a subscript?

k. The right hand side of the equation r = … on line 181 is not legible

l. Variables in Section 3.4 also lack definitions and any understanding. As a result, it is hard to evaluate the merit of the proposed method or determine what the proposed operations are doing.

m. The sentences between lines 187 and 188 are not clear.

n. The variables in Section 3.5 also lack definitions and any sense of intuition

o. Is U_{ad}, on line 197, the Cartesian product of Sobolev spaces? Or are the x variables in line 198 actually x’s. Please clarify.

p. It would be nice to see on or near line 201 that this is essentially the equations of linear elasticity and lambda and mu are Lame parameters.

q. Again, Equations 24 through 27 are not readable and lack any intuition. Where are these coming from? What do they mean? Each in isolation can be digested, but when all of them are crowded into a few very small lines, they remain indecipherable.

r. What are the variables used in Equations 28 and 29? Only some of them are declared in the subsequent lines. What about alpha? \bar{v} (Poisson’s ratio?) Please clarify. It is also hard to see which parentheses lead to multiplications and which ones are function evaluations. Could you use a dot product dot to indicate multiplication?

s. Lines 218 through 248 again lack any intuition. From what equations are these equations substituted into? Again, a lack of understanding, intuition, and definitions on most of the variables used herein make this feel more like spewing a list of equations rather than doing anything to build intuition, motivation, or confidence in the methods used.

t. What kinds of isoparametric Lagrangian symplectic triangle finite elements are used? Quadratic?

u. It appears that some variables may be overloaded (i.e. used twice to define different phenomena), such as A in line 141 and in line 179. These overloads should be removed---particularly because there is already a significant amount of different information being conveyed.

v. The Gauss tensor, Q_{\alpha \beta}, appears to be first introduced in line 375 of the document. Please introduce earlier and define earlier.

2. Some of the symbols defined appear to never be used, such as the Christoffel symbol. Please review the article to better clarify symbols used and to remove any extraneous symbols.

3. The article currently feels like a compilation of many different methods into a single method. However, the choice of these methods is not motivated. Do the assumptions of these methods work in conjunction with each other? Why, among all methods, were these methods chosen? Is there novelty beyond simply putting everything together? If so, what is it?

4. Why are different specimen dimensions used for shrinkage and drying experiments in Section 4.2.2? Were both following a methodology published in the literature? If not, why the variation?

5. What variables are fit in lines 301 and 302? Are these corresponding to Equations 3 and 5? If so please clarify.

6. How was the time evolution of water content measured in Section 5.1.ii? Please also clarify how measurements were taken and the meaning of the quantities in the legend of Figure 5.

7. How were damage evolution parameters determined? Are they based on experimental data, or on numerical simulations, based on reference from other publications?

8. There is not a clear distinction between what measurements and data relate to experimental data and what relates to numerical data throughout Section 5.2. Please clarify.

9. Many of the figures show behavior that has not leveled off asymptotically, such as in Figure 12, Figure 13, and Figures 14. These are for structural information that should stabilize. Why is such a stabilization not present? Is this due to lack of computing resources, or ability to measure? I question the validity of the model if, for example, the thickness of the shell linearly deteriorates indefinitely, or if the strain at node E continues to change indefinitely.

10. Some of the labels and graphics are difficult to read because lines are present on top of other lines. If this is the case, could the authors please put nodes that correspond to different graph lines in different locations so figures are more easily decipherable?

11. It is not clear if there was any validation of the model on the actual constructed structure. If not, please clarify why. For such a complex model, there should be some validation that the results produced are meaningful and give higher-fidelity results than significantly simpler models.

In addition, there are a number of typographical errors throughout the article (e.g. Finite Element Method should all be lower-case in the abstract, ThermoChemical and HydroMechanical should also be lower-case and hyphenated, the mapping u from the closure of the domain should probably go to R^3, etc). Overall, this did not impede my ability to read and follow the article, but removing the typos would be important prior to publication. Also, why is the Reissner-Mindlin shell element referred to as Q4GG? If this is particular notation to some software, please reference it or remove it. Otherwise, please cite the specific version of the Reissner-Mindlin shell that is the Q4GG element used.

Finally, some of the superscripts and subscripts on symbols---particularly those using a percent sign as a subscript----were difficult to read and should be typeset in a more readable manner.

6. PLOS authors have the option to publish the peer review history of their article (what does this mean? ). If published, this will include your full peer review and any attached files.

**Do you want your identity to be public for this peer review?** For information about this choice, including consent withdrawal, please see our Privacy Policy .

Reviewer #1: No

Reviewer #2: No

Reviewer #3: No

---

## [Author Response · Author response to Decision Letter 1]

18 Dec 2024

Reviewer #1: • Overall Strengths: The manuscript offers a novel contribution to the field by integrating multiple phenomena that affect concrete performance over time, making it particularly valuable for long-term structural design applications. The proposed TCHM model is robust and well-supported by experimental data.

• Areas for Improvement:

1. Data Interpretation: Provide more detailed explanations for intermediate results, particularly regarding the effects of hydration and delayed deformations on the overall structural behavior.

Answer : Details provided see lines 315-318 and 338-340

2. Statistical Analysis: Consider performing sensitivity analysis to further validate your model and provide deeper insights into the impact of key parameters.

Answer : Please see table 3.

3. Data Sharing: Ensure that all underlying data, including raw measurements, are made fully available in accordance with PLOS ONE's data policy.

Answer : The data may be available upon formal request to the project owner, as specified in the initial submission.

4. Language: Simplify the presentation of some technical content to improve readability and correct minor typographical and grammatical errors.

Answer : A proofreading was done and the typos corrected.

Conclusion: The manuscript is scientifically sound and offers important insights for concrete shell design. With minor revisions to improve data accessibility, statistical rigor, and clarity, it is suitable for publication.

Reviewer #2: The paper covers an important topic, but the authors need to strengthen the logical reasoning and clarity of the results. Specifically:

1. Revise the abstract to include details about the experimental work, outcomes, and recommendations.

Answer : Details provided see lines 19-22

2. Provide more in-depth explanations in the section discussing linear shrinkage.

Answer : Details provided see lines 332-334

3. Add a paragraph outlining potential future research directions based on the current study.

Answer : Details provided see lines 405-410

4. Include a paragraph at the end of the introduction emphasizing the research significance of the study.

Answer : Details provided see lines 112-114

5. Update the references with more recent research, as there are no citations from 2024 and only two from 2023, to better support the ideas and highlight the novelty of the work.

Answer : See [13], [21] and [40].

Reviewer #3: This article aims to describe a more complete framework by which to evaluate structural deformation in thin-shelled concrete dome structures: by including the contributions of thermochemical and hydromechanical behavior (e.g. creep) of concrete, in addition to typical elastic deformation. The paper then showcases the results of the proposed methodology by an analysis of a structure designed like an aircraft, particularly in the fuselage-like portion.

While the proposed article has potential, and appears to investigate an area of interest, the article lacks clarity in a number of areas. This makes the article difficult to evaluate from the perspective of novelty, completeness, and value of the proposed method. Particularly, the article needs to better address the following areas:

1. Many of the terms that are used are not defined, and not motivated. Particularly,

a. It is not clear why the Kronecker delta is only define don the midsurface.

Answer : corrected

b. What is \mu^\rho_\alpha, which is on the last line of the first page of the document? What does it physically correspond to?

Answer : μ_α^ρ=δ_α^ρ-zB_α^ρ

It’s the matrix which makes it possible to connect the covariant base of the 3D shell to the covariant base on the mid-surface.

c. Does the document use summation notation on indices or not?

Answer : In some relationship yes not in others.

d. What is the physical meaning of \psi (the first equation of the second page?

Answer : rate of change of volume. Addition added see line 35.

e. When \omega_L is introduced on page 5, it should be explicitly mentioned (using the language from the start of the text) that this is the free water content.

Answer : corrected

f. A description of the variables and parameters in Equations 2 and 3 are not present in the text. What are these? Without physical intuition and meaning, it is hard to motivate why the method of choice by the authors is appropriate.

Answer : description done see lines 148-151

g. What does “The drying parameter A and \delta are numerically adjusted after the resolution” mean (see line 141 of the text)? Are these parameters that are fit based on experimental data? Or to match a given deformation? Please clarify.

Answer : The clarification provided see lines 155-156

h. Is k of line 148 the same as \kappa in line 147? What do these calibration parameters correspond to? What is their physical meaning?

Answer : input error correction provided see lines 163-164

i. Variables in Section 3.3 lack definition, explanation, and intuition. Please label these variables and describe what they mean. What is the intuition that a practitioner can use to understand these? Also, for equation 13, what is the meaning of the numbers 1000 for concretes with silica fume and 3200 for those without silica fume? What is the repercussion for changing these numbers in this way? Where did these numbers come from?

Answer : details provided in section 3

j. Should Ac and Bc of equations 179 and 180 be A_c and B_c, where the _ denotes a subscript?

Answer : The clarification provided see lines 186-191

k. The right hand side of the equation r = … on line 181 is not legible

Answer : modification provided see line 199

l. Variables in Section 3.4 also lack definitions and any understanding. As a result, it is hard to evaluate the merit of the proposed method or determine what the proposed operations are doing.

Answer : comments provided see line 179-181

m. The sentences between lines 187 and 188 are not clear.

Answer : modification provided see line 199

n. The variables in Section 3.5 also lack definitions and any sense of intuition

Answer : clarification provided

o. Is U_{ad}, on line 197, the Cartesian product of Sobolev spaces? Or are the x variables in line 198 actually x’s. Please clarify.

Answer : clarification provided, see also line 28

p. It would be nice to see on or near line 201 that this is essentially the equations of linear elasticity and lambda and mu are Lame parameters.

Answer : clarification provided

q. Again, Equations 24 through 27 are not readable and lack any intuition. Where are these coming from? What do they mean? Each in isolation can be digested, but when all of them are crowded into a few very small lines, they remain indecipherable.

Answer : clarification provided

r. What are the variables used in Equations 28 and 29? Only some of them are declared in the subsequent lines. What about alpha? \bar{v} (Poisson’s ratio?) Please clarify. It is also hard to see which parentheses lead to multiplications and which ones are function evaluations. Could you use a dot product dot to indicate multiplication?

Answer : clarification provided

s. Lines 218 through 248 again lack any intuition. From what equations are these equations substituted into? Again, a lack of understanding, intuition, and definitions on most of the variables used herein make this feel more like spewing a list of equations rather than doing anything to build intuition, motivation, or confidence in the methods used.

Answer : After the variational formulation we reorganise the equation by grouping similar terms in order to facilitate numerical implementation in the FEM code.

t. What kinds of isoparametric Lagrangian symplectic triangle finite elements are used? Quadratic?

Answer :Only Triangle elements are used in the mesh with Quadratic polynomials as shape functions for tranverse displacements and linear for the others

u. It appears that some variables may be overloaded (i.e. used twice to define different phenomena), such as A in line 141 and in line 179. These overloads should be removed---particularly because there is already a significant amount of different information being conveyed.

Answer : correction provided

v. The Gauss tensor, Q_{\alpha \beta}, appears to be first introduced in line 375 of the document. Please introduce earlier and define earlier.

Answer : correction provided see lines 276-277

2. Some of the symbols defined appear to never be used, such as the Christoffel symbol. Please review the article to better clarify symbols used and to remove any extraneous symbols.

Answer : Christoffel symbol is used see line 273

3. The article currently feels like a compilation of many different methods into a single method. However, the choice of these methods is not motivated. Do the assumptions of these methods work in conjunction with each other? Why, among all methods, were these methods chosen? Is there novelty beyond simply putting everything together? If so, what is it?

Answer : It’s not a compilation of methods . Unlike in earlier works in this field where the different phenomenon were considered seperately. In reality they are concomitant and should be considered dependent in a realistic analysis. Herein, the complex coupled equations are considered and numerical calculations have been successfully implemented on the model. Figure 2 presents the skeleton of the proposed model.

4. Why are different specimen dimensions used for shrinkage and drying experiments in Section 4.2.2? Were both following a methodology published in the literature? If not, why the variation?

Answer : Yes both dimensions were used in some publications in the litterature

5. What variables are fit in lines 301 and 302? Are these corresponding to Equations 3 and 5? If so please clarify.

Answer : clarification provided

6. How was the time evolution of water content measured in Section 5.1.ii? Please also clarify how measurements were taken and the meaning of the quantities in the legend of Figure 5.

Answer : clarification provided see lines 351-354

7. How were damage evolution parameters determined? Are they based on experimental data, or on numerical simulations, based on reference from other publications?

Answer : clarification provided, based on references.

8. There is not a clear distinction between what measurements and data relate to experimental data and what relates to numerical data throughout Section 5.2. Please clarify.

Answer : In section 5.2 this concerns the presentation of the simulation results, recalling the useful data where needed.

9. Many of the figures show behavior that has not leveled off asymptotically, such as in Figure 12, Figure 13, and Figures 14. These are for structural information that should stabilize. Why is such a stabilization not present? Is this due to lack of computing resources, or ability to measure? I question the validity of the model if, for example, the thickness of the shell linearly deteriorates indefinitely, or if the strain at node E continues to change indefinitely.

Answer : we observe quasi-asymptotic behavior, The simulations are carried out over 50 days. We remain convinced based on the trends that asymptotic behavior will occur. The available resources took us 8 days for a compilation with this spatial and temporal mesh step.

10. Some of the labels and graphics are difficult to read because lines are present on top of other lines. If this is the case, could the authors please put nodes that correspond to different graph lines in different locations so figures are more easily decipherable?

Answer : clarifications provided.

11. It is not clear if there was any validation of the model on the actual constructed structure. If not, please clarify why. For such a complex model, there should be some validation that the results produced are meaningful and give higher-fidelity results than significantly simpler models.

Answer : Yes, on the basis of project data (architectural design, concrete mixture and climatlc conditions) we reproduced the experiment to carry out the various calibrations. The finite element validation was tested on several structures (real and benchmark in the literature [40-41])

In addition, there are a number of typographical errors throughout the article (e.g. Finite Element Method should all be lower-case in the abstract, ThermoChemical and HydroMechanical should also be lower-case and hyphenated, the mapping u from the closure of the domain should probably go to R^3, etc). Overall, this did not impede my ability to read and follow the article, but removing the typos would be important prior to publication. Also, why is the Reissner-Mindlin shell element referred to as Q4GG? If this is particular notation to some software, please reference it or remove it. Otherwise, please cite the specific version of the Reissner-Mindlin shell that is the Q4GG element used.

Answer : clarifications provided.

Finally, some of the superscripts and subscripts on symbols---particularly those using a percent sign as a subscript----were difficult to read and should be typeset in a more readable manner.

Answer : clarifications provided.

---

## [Decision Letter · Decision Letter 1]

PONE-D-24-21940R1

ANALYSIS OF A SELF-SUPPORTING SHELL CONCRETE ROOF WITH NONLINEAR COUPLED EVOLUTIVE MATERIAL PARAMETERS.

PLOS ONE

Dear Dr. DJOPKOP,

Thank you for submitting your manuscript to PLOS ONE. After careful consideration, we have decided that your manuscript does not meet our criteria for publication and must therefore be rejected.

Specifically:

**see below **

I am sorry that we cannot be more positive on this occasion, but hope that you appreciate the reasons for this decision.

Kind regards,

Pawel Klosowski, D.Sc.

Academic Editor

PLOS ONE

Additional Editor Comments:

The previous version of the paper has one Author, the current one has five additional co-authors. In the cover letter there is no explanation. It is not ethic to add coauthors when one expect that paper will be published. Therefore I decided to reject the paper.

Reviewers' comments:

Reviewer's Responses to Questions

**Comments to the Author**

1. If the authors have adequately addressed your comments raised in a previous round of review and you feel that this manuscript is now acceptable for publication, you may indicate that here to bypass the “Comments to the Author” section, enter your conflict of interest statement in the “Confidential to Editor” section, and submit your "Accept" recommendation.

Reviewer #2: All comments have been addressed

2. Is the manuscript technically sound, and do the data support the conclusions?

Reviewer #2: Partly

3. Has the statistical analysis been performed appropriately and rigorously? 

Reviewer #2: Yes

4. Have the authors made all data underlying the findings in their manuscript fully available?

Reviewer #2: Yes

5. Is the manuscript presented in an intelligible fashion and written in standard English?

Reviewer #2: No

6. Review Comments to the Author

Reviewer #2: There are some grammatical mistakes in the manuscript that need to be removed such as wrong spelling of thickness in abstract.

7. PLOS authors have the option to publish the peer review history of their article (what does this mean? ). If published, this will include your full peer review and any attached files.

**Do you want your identity to be public for this peer review?** For information about this choice, including consent withdrawal, please see our Privacy Policy .

Reviewer #2: **Yes: ** Asad Zia

- - - - -

---

## [Author Response · Author response to Decision Letter 2]

20 Jan 2025

Another complete proofreading was done and typos pruned

---

## [Decision Letter · Decision Letter 2]

PONE-D-24-21940R2ANALYSIS OF A SELF-SUPPORTING SHELL CONCRETE ROOF WITH NONLINEAR COUPLED EVOLUTIVE MATERIAL PARAMETERS.PLOS ONE

Dear Dr. LANDRY KOUANANG DJOPKOP,

Thank you for submitting your manuscript to PLOS ONE. After careful consideration, we feel that it has merit but does not fully meet PLOS ONE’s publication criteria as it currently stands. Therefore, we invite you to submit a revised version of the manuscript that addresses the points raised during the review process.

We look forward to receiving your revised manuscript.

Kind regards,

Dajiang Geng

Academic Editor

PLOS ONE

1. We notice that your manuscript file was uploaded on December 14, 2024. Please can you upload the latest version of your revised manuscript as the main article file, ensuring that does not contain any tracked changes or highlighting. This will be used in the production process if your manuscript is accepted. Please follow this link for more information: http://blogs.PLOS.org/everyone/2011/05/10/how-to-submit-your-revised-manuscript/

Additional Editor Comments (if provided):

Reviewers' comments:

Reviewer's Responses to Questions

**Comments to the Author**

1. If the authors have adequately addressed your comments raised in a previous round of review and you feel that this manuscript is now acceptable for publication, you may indicate that here to bypass the “Comments to the Author” section, enter your conflict of interest statement in the “Confidential to Editor” section, and submit your "Accept" recommendation.

Reviewer #3: (No Response)

Reviewer #4: (No Response)

2. Is the manuscript technically sound, and do the data support the conclusions?

Reviewer #3: Partly

Reviewer #4: Partly

3. Has the statistical analysis been performed appropriately and rigorously? 

Reviewer #3: Yes

Reviewer #4: Yes

4. Have the authors made all data underlying the findings in their manuscript fully available?

Reviewer #3: Yes

Reviewer #4: No

5. Is the manuscript presented in an intelligible fashion and written in standard English?

Reviewer #3: No

Reviewer #4: No

6. Review Comments to the Author

Reviewer #3: Review of

ANALYSIS OF A SELF-SUPPORTING SHELL CONCRETE ROOF WITH NONLINEAR COUPLED EVOLUTIVE MATERIAL PARAMETERS

The quality of the manuscript has improved significantly with the most recent revision. Many of the ideas that were unclear are now clear. However, there are still a number of remaining concerns.

1. Please make \mu^\rho_\alpha into its own equation shortly before or after line 32.

2. There is a host of variables that are used herein. However, the nomenclature section only contains about ½ of them. Please place all variables in the nomenclature section and define them therein so that readers know at least one place they can go to follow what is being written. Current in-text definitions are also helpful, but are insufficient (as discussed more below).

3. From what I can see, summation notation is used throughout. Please indicate this near the beginning of the document. If there are instances where summation notation is not used, please explicitly call these out.

4. Please clarify on Equation 12 if \epsilon_{cd} is a computed quantity or if it is provided from experiments. If it is computed (and leads to total strain), consider moving it to the left hand side of an equation.

5. The role of creep (Section 3.3) in the overall model. \epsilon^{cr} is defined in the beginning and used in Figure 2, but it does not seem to be present in Section 3.3. Additionally, the total creep function (Equation 14) is confusing in the following ways

a. The expression J(t,t_0) does not appear to be used anywhere else in the document. As a consequence, it is not clear how this contributes to creep (e.g. in the algorithm of Figure 2). Please clarify. How is this related to \epsilon^{cr}?

b. The quantities E_c, E_{c28}, and s in Equation 14 are not defined.

c. Both the self-creep and the desiccation creep are defined (in Equations 9 and 12, respectively) prior to their use in Equation 14. However, the other term (1/E_c(t_0)) is never discussed prior to its use in Equation 14. What does this represent?

d. Why are the creep terms divided by the E_{c28} terms?

6. Quantities in Section 3.4 are undefined and often not motivated.

a. Particularly, what do Y_0, Y, A, B, \epsilon_do \epsilon, A_t, j, k, r, A_c, B, B_t, B_c, \tilde{sigma_i}, \epsilon_{eq}^{\corr}, <__>_{-}, <__>_{+}, and \gamma physically mean?

b. Where does someone trying to reproduce this work get information about how to get these variables? A couple are discussed as equations (e.g. Y_0), but others (e.g. k) are not. Please be clearer. The final paragraph of Section 3.4 is a good start, but is insufficient.

7. Quantities in Section 3.5 also need clarification.

a. Is \bar{q} a number? If so, why does it does it have superscripts in line 282? If not, why is the set of admissible displacements only the Cartesian product of 4 Sobolev spaces (i.e. 3 for u and 1 for q)?

b. Please describe what the stretching displacement does in words.

c. S (in H^1(S)) is not defined. Is this the surface? The midsurface? The entire domain? It appears to be defined as the midsurface in Line 232, but it is first used in Line 224. This is confusing.

d. G^{\alpha \beta} (and similar indexed terms on e.g. Line 229) are never defined, though it is presumed that they relate to the contravariant bases of the shell. Are they inner products of these bases? Or subscripts of the bases? Please clarify.

e. A in Line 229 is not defined, and its relationship with C in Line 234 is also unclear. Please clarify.

f. Subscript indices on \epsilon^\eta and \epsilon^q in line 238 appear to be missing.

g. Tensors e, k, and Q are used in Equation 26 (Line 241), but are undefined. They appear to be defined on Lines 275, 277, and 278. Please define these shortly after their first use, rather than defining them over an entire page later.

h. The tensor Y on line 260 is not defined. Please clarify.

i. What are the terms in the linear form (i.e. L(v,\bar{y}) on line 282? Similarly, what are the terms in the integrands of Line 283. Define all terms so that the information can be used and so that it is appropriately motivated. Define them both using equations (where relevant) and verbally. If it is a linear form, why is q (an unknown as per Line 218) present in it?

j. Notation with \partial_\alpha (e.g. line 261) and with commas in index notation (e.g. line 276) are both used. Please clarify if both are partial derivatives or if they indicate different behavior.

8. Figure 2 is the only way to understand how all of the information from Section 3 is integrated. However, it is not well introduced in the text of Section 4. Please spend more time discussing this figure so that readers can understand how the information from Section 3 can be used in your model. Also, there appear to be many, many more input parameters for the algorithm than just the time of simulation and the number of time increments (e.g. Lame parameters and/or Young’s modulus and Poisson’s ratio, j=0.7, etc).

9. Please improve Figure 2 to

a. Include arrows between every box/shape.

b. Have a caption that more fully describes the algorithm.

c. If the numerical experiment is only a maximum of 50 days, why is the creep strain ignored until after 28 days? Please clarify in the text.

d. It appears that the algorithm is nonlinear. However, the nonlinearity is not clearly indicated (through e.g. arrows) in Figure 2. Please clarify the iterative/nonlinear process for each time step, or please clarify that the method is not linear.

10. The numerical implementation section spends a significant amount of discussion on how parameters were extracted using experiments. Please separate the experimental/material parameter discussion into a separate section before the numerical implementation. This will also clarify what experimental work is done and what is actually numerical.

11. Similarly, the results and discussions section makes it unclear what results are numerical and what results are due to the model. Please explicitly declare what results are experimental and which ones are numerical. It is my understanding that almost all results are numerical except for those in the “Intermediate Results” section. Is this correct?

12. The text between Lines 422 and 427 confuses me. Does this indicate that for a static (elastic) analysis to produce the same amount of displacement as this model (which includes creep, damage, thermal effects, etc), the static analysis would have to apply a force 183x larger than that applied when including creep, damage, etc?

13. Temperatures on each day are shown in Figure 8. However, analysis was only performed for 50 days. On what day did this analysis begin? Which temperature values were used, and over which of 50 days of the 365 displayed?

14. What time steps were used for this analysis? One time step per day? What kind of solver was used? Implicit or explicit?

15. Please clarify in the text why you can be confident that the chosen time step increments were appropriate and stable. Similarly, if there is a nonlinearity in each time step, please indicate why you can have confidence that the number of nonlinear iterations was appropriate to resolve behavior in each time step.

16. The authors indicate that they believe that their simulations approach asymptotic behavior. However, Figures 12, 13, and 14 do not seem to demonstrate that the model reaches an asymptotic stability over the course of 50 days. In the rebuttal, authors have declared that they lack computational ability to perform additional analysis and that this one already took 8 days. This makes sense. However, the authors should clarify in the conclusions that asymptotic behavior for their model is believed, but that further research is needed to clarify that the proposed method is applicable for long timeframes. Particularly, it is not clear to the reviewer that displacement will not increase indefinitely and that the model accurately represents what would be anticipated experimentally. This deficiency does not discredit this work; however, this lack of clarity should be openly stated.

17. Readability issues:

a. Please revisit the sentence starting on line 86 for clarity.

b. Line 169: “Given the small size of the shell…” Should this be “Given the small thickness of the shell…”?

c. The sentence beginning on line 190 is incomplete.

d. Please indicate units of liters using a capital L rather than a lower-case L.

Reviewer #4: This paper considers the thermochemical and hydro-mechanical (TCHM) behavior of

Concrete in the analysis. Overall, this is a fascinating topic. My main concern is that it is unclear how many specimens were used in the experimental test to determine the concrete characteristics over time (shrinkage, deflection, thickness). It is not proper to conclude based on the few specimens.

This is an intriguing topic, and the overall approach is valid; I suggest it be considered for publishing after careful language editing and resolving the issues mentioned in the attachment.

7. PLOS authors have the option to publish the peer review history of their article (what does this mean? ). If published, this will include your full peer review and any attached files.

**Do you want your identity to be public for this peer review?** For information about this choice, including consent withdrawal, please see our Privacy Policy .

Reviewer #3: No

Reviewer #4: No

---

## [Author Response · Author response to Decision Letter 3]

19 Apr 2025

The quality of the manuscript has improved significantly with the most recent revision. Many of the ideas that were unclear are now clear. However, there are still a number of remaining concerns.

1. Please make \mu^\rho_\alpha into its own equation shortly before or after line 32.

Answer : done

2. There is a host of variables that are used herein. However, the nomenclature section only contains about ½ of them. Please place all variables in the nomenclature section and define them therein so that readers know at least one place they can go to follow what is being written. Current in-text definitions are also helpful, but are insufficient (as discussed more below).

Answer : It was pruned

3. From what I can see, summation notation is used throughout. Please indicate this near the beginning of the document. If there are instances where summation notation is not used, please explicitly call these out.

Answer : Specified note

4. Please clarify on Equation 12 if \epsilon_{cd} is a computed quantity or if it is provided from experiments. If it is computed (and leads to total strain), consider moving it to the left hand side of an equation.

Answer : is a computed quantity,

5. The role of creep (Section 3.3) in the overall model. \epsilon^{cr} is defined in the beginning and used in Figure 2, but it does not seem to be present in Section 3.3. Additionally, the total creep function (Equation 14) is confusing in the following ways

a. The expression J(t,t_0) does not appear to be used anywhere else in the document. As a consequence, it is not clear how this contributes to creep (e.g. in the algorithm of Figure 2). Please clarify. How is this related to \epsilon^{cr}?

Answer : clarification given

b. The quantities E_c, E_{c28}, and s in Equation 14 are not defined.

Answer : clarification given

c. Both the self-creep and the desiccation creep are defined (in Equations 9 and 12, respectively) prior to their use in Equation 14. However, the other term (1/E_c(t_0)) is never discussed prior to its use in Equation 14. What does this represent?

Answer : clarification given

d. Why are the creep terms divided by the E_{c28} terms?

Answer : relatiobship reported in [38]

6. Quantities in Section 3.4 are undefined and often not motivated.

a. Particularly, what do Y_0, Y, A, B, \epsilon_do \epsilon, A_t, j, k, r, A_c, B, B_t, B_c, \tilde{sigma_i}, \epsilon_{eq}^{\corr}, <__>_{-}, <__>_{+}, and \gamma physically mean?

Answer : clarification made in 3.4

b. Where does someone trying to reproduce this work get information about how to get these variables? A couple are discussed as equations (e.g. Y_0), but others (e.g. k) are not. Please be clearer. The final paragraph of Section 3.4 is a good start, but is insufficient.

Answer : we have enriched and the computed values are specified in section 5.2.1

7. Quantities in Section 3.5 also need clarification.

a. Is \bar{q} a number? If so, why does it does it have superscripts in line 282? If not, why is the set of admissible displacements only the Cartesian product of 4 Sobolev spaces (i.e. 3 for u and 1 for q)?

Answer : clarification provide between lines 245-250

b. Please describe what the stretching displacement does in words.

Answer : stretching displacement allows the calculation of shear strain and stress.

c. S (in H^1(S)) is not defined. Is this the surface? The midsurface? The entire domain? It appears to be defined as the midsurface in Line 232, but it is first used in Line 224. This is confusing.

Answer : clarification provide : S is the midsurface see line 257

d. G^{\alpha \beta} (and similar indexed terms on e.g. Line 229) are never defined, though it is presumed that they relate to the contravariant bases of the shell. Are they inner products of these bases? Or subscripts of the bases? Please clarify.

Answer : clarification made : see line 261

e. A in Line 229 is not defined, and its relationship with C in Line 234 is also unclear. Please clarify.

Answer : clarification made : see line 259

f. Subscript indices on \epsilon^\eta and \epsilon^q in line 238 appear to be missing.

Answer : it’s correct , this is another notation

g. Tensors e, k, and Q are used in Equation 26 (Line 241), but are undefined. They appear to be defined on Lines 275, 277, and 278. Please define these shortly after their first use, rather than defining them over an entire page later.

Answer : clarification made between lines 271-277

h. The tensor Y on line 260 is not defined. Please clarify.

Answer : correction made see line 297

i. What are the terms in the linear form (i.e. L(v,\bar{y}) on line 282? Similarly, what are the terms in the integrands of Line 283. Define all terms so that the information can be used and so that it is appropriately motivated. Define them both using equations (where relevant) and verbally. If it is a linear form, why is q (an unknown as per Line 218) present in it?

Answer : clarification made

j. Notation with \partial_\alpha (e.g. line 261) and with commas in index notation (e.g. line 276) are both used. Please clarify if both are partial derivatives or if they indicate different behavior.

Answer : clarification made see line 306

8. Figure 2 is the only way to understand how all of the information from Section 3 is integrated. However, it is not well introduced in the text of Section 4. Please spend more time discussing this figure so that readers can understand how the information from Section 3 can be used in your model. Also, there appear to be many, many more input parameters for the algorithm than just the time of simulation and the number of time increments (e.g. Lame parameters and/or Young’s modulus and Poisson’s ratio, j=0.7, etc).

Answer : details provided

9. Please improve Figure 2 to

a. Include arrows between every box/shape.

Answer : The arrows have been added

b. Have a caption that more fully describes the algorithm.

Answer : done

c. If the numerical experiment is only a maximum of 50 days, why is the creep strain ignored until after 28 days? Please clarify in the text.

Answer : clarification provided

d. It appears that the algorithm is nonlinear. However, the nonlinearity is not clearly indicated (through e.g. arrows) in Figure 2. Please clarify the iterative/nonlinear process for each time step, or please clarify that the method is not linear.

Answer : clarification provided

10. The numerical implementation section spends a significant amount of discussion on how parameters were extracted using experiments. Please separate the experimental/material parameter discussion into a separate section before the numerical implementation. This will also clarify what experimental work is done and what is actually numerical.

Answer : It wil be difficult for us to separate the results because it involves both experimental and numerical work. The experimental results allow us to calibrate the parameters for the proposed model. The results were presented in order.

11. Similarly, the results and discussions section makes it unclear what results are numerical and what results are due to the model. Please explicitly declare what results are experimental and which ones are numerical. It is my understanding that almost all results are numerical except for those in the “Intermediate Results” section. Is this correct?

Answer : clarification provided see lines 415-416

12. The text between Lines 422 and 427 confuses me. Does this indicate that for a static (elastic) analysis to produce the same amount of displacement as this model (which includes creep, damage, thermal effects, etc), the static analysis would have to apply a force 183x larger than that applied when including creep, damage, etc?

Answer : we are trying to show that nor taking into account the delayed effects, therefore the specificities of concrete, in the calculation phase would be a dangerous decision.

13. Temperatures on each day are shown in Figure 8. However, analysis was only performed for 50 days. On what day did this analysis begin? Which temperature values were used, and over which of 50 days of the 365 displayed?

Answer : we use 50 days, see lines 448-449.

14. What time steps were used for this analysis? One time step per day? What kind of solver was used? Implicit or explicit?

Answer : see lines 458-459.

15. Please clarify in the text why you can be confident that the chosen time step increments were appropriate and stable. Similarly, if there is a nonlinearity in each time step, please indicate why you can have confidence that the number of nonlinear iterations was appropriate to resolve behavior in each time step.

Answer : We did not use a resolution scheme for time discretization, we directly calculate the value at a given time. For each step we use the algorithm in figure 2

16. The authors indicate that they believe that their simulations approach asymptotic behavior. However, Figures 12, 13, and 14 do not seem to demonstrate that the model reaches an asymptotic stability over the course of 50 days. In the rebuttal, authors have declared that they lack computational ability to perform additional analysis and that this one already took 8 days. This makes sense. However, the authors should clarify in the conclusions that asymptotic behavior for their model is believed, but that further research is needed to clarify that the proposed method is applicable for long timeframes. Particularly, it is not clear to the reviewer that displacement will not increase indefinitely and that the model accurately represents what would be anticipated experimentally. This deficiency does not discredit this work; however, this lack of clarity should be openly stated.

Answer : In all the documents consulted, therefore some in reference which deal with a similar theme have an asymptotic behavior , also our results show us a creep dominance therefore the curve will be a asymptotic given the shape of the delay function.

17. Readability issues:

a. Please revisit the sentence starting on line 86 for clarity.

Answer : clarification provided, see lines 90-92

b. Line 169: “Given the small size of the shell…” Should this be “Given the small thickness of the shell…”?

Answer : corrected, see 172

c. The sentence beginning on line 190 is incomplete.

Answer : We have completed see lines 203-207

d. Please indicate units of liters using a capital L rather than a lower-case L.

Answer : correction provided

Reviewer #4: This paper considers the thermochemical and hydro-mechanical (TCHM) behavior of

Concrete in the analysis. Overall, this is a fascinating topic. My main concern is that it is unclear how many specimens were used in the experimental test to determine the concrete characteristics over time (shrinkage, deflection, thickness). It is not proper to conclude based on the few specimens.

This is an intriguing topic, and the overall approach is valid; I suggest it be considered for publishing after careful language editing and resolving the issues mentioned in the attachment.

Area to improve

Major:

1.1. It is not clear how many specimens were used in the experimental test to determine the concrete characteristics over time (shrinkage, deflection, thickness). It is not true to conclude based on the few specimens.

Answer : we have used 12 specimens in total

1.2. Determine how many cubic specimens were used to determine the compressive strength of concrete. I suggest a table or chart be presented to determine each specimen's strength and how its value is related to the strength of a ‘standard specimen’ (cylindrical specimens) according to the code (e.g., ACI).

Answer : Three test specimens were removed from the immersion ank on the 28th day and sent to an approved laboratory. This laboratory carried out the crushing in accordance with the NF P 18-406 standard prescribed by us.

1.3. This section (4.2.2.) requires more clarification; probably, adding some more figures and diagrams will probably help.

Answer : corrected

1.4. The details of the FEA model are not provided.

1.5. The accuracy of the finite element model should be evaluated and then applied to this specific case study.

Answer : it’s done in previous work [40-41]

Minor:

2. Quality of writing

2.1. This manuscript requires significant language editing. There are many broken sentences. It makes it hard for the reader to understand it because of the poor writing; however, this is an interesting topic, and I would suggest it for publication after careful language editing.

Answer : The corrections have been provided

2.2. I suggest the authors names be mentioned in the text, which increases the readability. For example: instead of Later, [7] proposed … Later, Borack [7] proposed.

Answer : done

2.3. Passive language is more effective and common in scientific papers; please revise it carefully. For example, in the abstract, we showed… change to: results indicated that… Or on page 4, We begin by describing…

Answer : revisions have been done

3. Novelty, problem statement

3.1. Well identified

3.2. The research question is clearly stated

3.3. Essential literature was mentioned; however, the references are not very recent. More recent works should be considered (2022-25).

4. Title, Abstract, keywords

4.1. Add more general and specific keywords to help your work be identified more easily by potential readers (e.g., some keywords for containing shell structures, NONLINEAR COUPLED EVOLUTIVE MATERIAL, and thermochemical and hydro-mechanical (TCHM) behavior of concrete)

Answer : Keywords added

4.2. Passive language is more effective and common in scientific papers; please revise it carefully. For example, in the abstract, we showed… change to: results indicated that…

Answer : corrections provided

4.3. The abstract doesn’t cover the method explicitly used. Please add a few sentences. Additionally, the experimental test to provide the data should be included.

Answer : a few sentences have been added

5. Introduction

5.1. Define very briefly (2-3 sentences) what ‘NONLINEAR COUPLED EVOLUTIVE MATERIAL’ means in this study in the introduction.

Answer : Firstly, we have internal dissipation, which is linked to phenomena such as shrinkage, creep, etc. These independent specificities will give concrete non-linear behavior. We will therefore have material or rheological non-linearity.

6. Method

6.1. The equation provided on page 4, line 126, doesn’t have a number. Additionally, provide additional information on how (b) is determined in this formulation.

Answer : Corrections made

6.2. Page 7, line 191: Recommended value of k = 0.7; this should be backed by a reference.

Answer : Correction made

6.3. In many cases, there is a lack of information/description of the variables and constants in the formulas. (e.g., Eq19, Eq37, Eq 24, …).

Answer : Corrections made

7. Results

7.1. Add some description (maximum a paragraph) in the manuscript on ‘Calculation algorithm implementing the proposed model’ presented in Figure 2. It is not clear what happens in this algorithm. The left part (read connection to the left line) appears to be redundant (?).

Answer : Clarification provided

7.2. Suggestion: A comparison of this shell while designed with conventional methods and when these defects are considered will be beneficial and add more weight to your work. (optional)

Answer : We present a comparison of lines 502-507

8. Conclusion

8.1. The conclusion requires more specific findings. It is too short and doesn’t contain all the findings. The main findings and conclusions and probable impact of the work could be mentioned here.

Answer : The conclusion was added

9. Figures, Tables, charts

9.1. In Table 1, determine if the water is in liters or kg?

Answer : correction done

10. Rewrite the author's contribution section by determining who has done what part of the work in detail. Please refer to the journal guidelines. (optional)

Answer : added

11. Most references are outdated; please mention and relate some recent publications. (2023-25

Answer : see [22], [40],

---

## [Editor Report · Decision Letter 3]

PONE-D-24-21940R3ANALYSIS OF A SELF-SUPPORTING SHELL CONCRETE ROOF WITH NONLINEAR COUPLED EVOLUTIVE MATERIAL PARAMETERS.PLOS One

Dear Dr. DJOPKOP,

Thank you for submitting your manuscript to PLOS One. After careful consideration, we feel that it has merit but does not fully meet PLOS One’s publication criteria as it currently stands. Therefore, we invite you to submit a revised version of the manuscript that addresses the points raised during the review process.

The Academic Editor has assessed your revisions and they are satisfied that all scientific concerns have been addressed. However, before we proceed with publication, we kindly ask you to thoroughly copyedit your manuscript for language usage, spelling, and grammar. If you do not know anyone who can help you do this, you may wish to consider employing a professional scientific editing service. The American Journal Experts (AJE) (https://www.aje.com/) is one such service that has extensive experience helping authors meet PLOS guidelines and can provide language editing, translation, manuscript formatting, and figure formatting to ensure your manuscript meets our submission guidelines. Please note that having the manuscript copyedited by AJE or any other editing services does not guarantee selection for peer review or acceptance for publication. Upon resubmission, please provide the following: 1) The name of the colleague or the details of the professional service that edited your manuscript

3) A clean copy of the edited manuscript (uploaded as the new *manuscript* file). We thank you for your attention to this request, and we look forward to receiving your revised manuscript.

If applicable, we recommend that you deposit your laboratory protocols in protocols.io to enhance the reproducibility of your results. Protocols.io assigns your protocol its own identifier (DOI) so that it can be cited independently in the future. For instructions see: https://journals.plos.org/plosone/s/submission-guidelines#loc-laboratory-protocols . Additionally, PLOS One offers an option for publishing peer-reviewed Lab Protocol articles, which describe protocols hosted on protocols.io. Read more information on sharing protocols at https://plos.org/protocols?utm_medium=editorial-email&utm_source=authorletters&utm_campaign=protocols .

We look forward to receiving your revised manuscript.

Kind regards,

Hugh Cowley

Senior Editor

PLOS One

on behalf of

Dajiang Geng

Academic Editor

PLOS One
---

## [Author Response · Author response to Decision Letter 4]

14 May 2025

Answer: the document was edited by AJE

The American Journal Experts (AJE) (https://www.aje.com/)

Answer: Done

Answer:Done

---

## [Editor Report · Decision Letter 4]

ANALYSIS OF A SELF-SUPPORTING SHELL CONCRETE ROOF WITH NONLINEAR COUPLED EVOLUTIVE MATERIAL PARAMETERS.

PONE-D-24-21940R4

Dear Dr. LANDRY KOUANANG DJOPKOP,

We’re pleased to inform you that your manuscript has been judged scientifically suitable for publication and will be formally accepted for publication once it meets all outstanding technical requirements.

Kind regards,

Dajiang Geng

Academic Editor

PLOS ONE
---

## [Editor Report · Acceptance letter]

PONE-D-24-21940R4

PLOS ONE

Dear Dr. KOUANANG,

I'm pleased to inform you that your manuscript has been deemed suitable for publication in PLOS ONE. Congratulations! Your manuscript is now being handed over to our production team.

Kind regards,

on behalf of

Dr. Dajiang Geng

Academic Editor

PLOS ONE